



# Microwave implementation of two-source energy balance approach for estimating evapotranspiration

Thomas R. Holmes[1], Christopher Hain[2], Wade T. Crow[3], Martha C. Anderson[3], William P. Kustas[3]

[1]Hydrologcal Sciences Lab, NASA Goddard Space Flight Center, Greenbelt, MD, 20771, USA
[2]Earth Science Office, NASA Marshall Space Flight Center, Huntsville, AL, USA
[3]Hydrology and Remote Sensing Laboratory, USDA-ARS, Beltsville, MD 20705, USA

*Correspondence to*: Thomas R. Holmes (thomas.r.holmes@nasa.gov)

**Abstract.** A newly developed microwave (MW) land surface temperature (LST) product is used to substitute thermal infrared (TIR) based LST in the two-source energy balance approach (TSEB) for estimating ET from space. This TSEB land surface scheme, the Atmosphere Land Exchange Inverse (ALEXI) model framework, is an approach that minimizes sensitivity to absolute biases in input records of LST through the analysis of the rate of temperature change in the morning. This experiment is therefore an important test of the ability to retrieve diurnal temperature information from a constellation of satellites with microwave radiometers that together provide 6-8 observations of Ka-band brightness temperature per location per day. This represents the first ever attempt at a global implementation of ALEXI with MW-based LST and is intended as the first step towards providing all-weather capability to the ALEXI framework. The leveraging of all sky capability of MW sensors is the main motivation of this work, as TIR-based ALEXI is limited to clear sky conditions.

The analysis is based on a 9-year long record of ALEXI ET generated with MW-LST as an input, which is compared to an existing implementation of the same framework with thermal infrared based LST. In this study, the MW-LST sampling is restricted to the same clear sky days as in the IR-based implementation to be able to analyse the impact of changing the LST dataset separately from the impact of sampling all-sky conditions. The results show that long-term bulk ET estimates agree with a spatial correlation of 92% for total ET in the Europe/Africa domain and agreement in seasonal (3-month) totals of 83-97 % depending on the time of year. Most importantly, the ALEXI-MW also matches ALEXI-IR very closely in terms of 3-month inter-annual anomalies, demonstrating its ability to capture the development and extent of drought conditions. The weekly ET output from the two parallel ALEXI implementations is further compared to a common ground measured reference provided by the FLUXNET consortium. Overall, they indicate a surprisingly close match in both performance metrics (correlation and RMSE) for all but the most challenging sites in terms of spatial heterogeneity and level of aridity. Moreover, merging MW- and IR-based ALEXI may provide estimates of ET with a reduced uncertainty, even during nominally clear sky days. It is concluded that a constellation of MW satellites can effectively be used to provide LST for estimating ET through TSEB, which is an important step towards all-sky satellite-based ET estimates.

## 1 Introduction

Estimating terrestrial evapotranspiration (ET) at continental to global scales is central to understanding the partitioning of energy and water at the earth surface and for evaluating modelled feedbacks operating between the atmosphere and biosphere. ET is an important flux that links the water, carbon, and energy cycles (Campbell and Norman, 1998). Approximately two-thirds of the precipitation over land is returned to the atmosphere by ET (Baumgartner & Reichel, 1975). Moreover, ET consumes 25-30% of the net radiation reaching the land surface (Trenberth et al., 2009). ET occurs as a result of atmospheric demand for water vapor and depends on the availability of water and energy. When plants are present, this balancing is controlled by leaf-level stomatal controls, and in agricultural areas the water availability may also be managed at the field scale through irrigation or drainage. The high spatial and temporal variability in the driving mechanisms in combination with possible field-scale management decisions poses a significant challenge to bottom-up modelling of ET at sub-monthly time scales, even at the spatial scales of numerical weather



prediction (NWP) models (5-25 km). In order for NWP models to improve the characterization of the surface energy budget, there is a need for timely diagnostic information on ET (Hain et al., 2015). This, in turn, could lead to a more timely and accurate identification of developing droughts (Anderson, 2011) which would aid farm-level management decisions.

ET is highly variable in space, so no amount of ground stations can provide an accurate estimate of the spatial average over larger domains, let alone the globe. Therefore, approaches have been developed to integrate satellite data with models to estimate ET from space. Surface energy balance approaches use surface temperature observations as the main diagnostic to estimate ET by partitioning the available energy into turbulent fluxes of sensible heating (H) and latent heating (LE). In the two source energy balance (TSEB) approach (Kustas and Norman, 1999; Norman et al., 1995) the partitioning is evaluated for the soil and the canopy separately. Anderson et al (1997) modified TSEB to leverage observations of the time evolution of surface temperature as a way to reduce the impact of biases in instantaneous temperature observations on the ET retrieval. This approach allowed for regional implementation of TSEB and came to be known as the Atmosphere Land Inverse Exchange (ALEXI) method (Anderson et al., 2007a; Mecikalski et al., 1999).

To date, ALEXI has always been implemented with land surface temperature (LST) retrievals from thermal infrared (TIR) imaging radiometers (Anderson et al., 2011). Most applications of ALEXI have utilized data products from geostationary satellites, for example the Geostationary Operational Environmental Satellite (GOES) with coverage over the Americas. More recently it has been applied to records from polar orbiting satellites to obtain consistent global coverage with short latency. This is based on day-night temperature differences from the Moderate Resolution Imaging Spectroradiometer (MODIS) on the Aqua satellite from NASA's Earth Observing System (EOS) program (Hain and Anderson, in review). Reliance on TIR effectively limits surface observations to clear skies (Rossow et al., 1989), and failure to completely mask cloud affected observations is shown to limit the precision in TIR-LST (Holmes et al., 2016). Continuous daily estimates of ET are generated from clear-sky ALEXI samples through temporal interpolation based on maintaining a normalized flux partitioning metric. In ALEXI this also accounts for daily evaporative losses (Anderson et al., 2007a). Recent work by Alfieri et al. (2017) analysed measurements from eddy-covariance towers and found the persistence for energy flux partitioning metrics to be short. In their analysis, they found that a return interval of no more than 5 days is necessary to keep the relative error in daily ET below 20 %.

In order to provide a more consistent and short return interval for daily ET measurements there is a need for accurate values during cloudy intervals. The approach we take here to address this challenge is to leverage passive microwave (MW) observations. The longer wavelengths (0.1-1 m) make MW observations of the land surface generally less susceptible to scattering and absorption by clouds than observations in the TIR spectral region (except for notable water and oxygen absorption windows; Ulaby et al. (1986)). One MW frequency band with a particularly high sensitivity to LST (Prigent et al., 2016) and high tolerance to clouds (Holmes et al., 2016) is Ka-band (36-37 GHz). MW radiometers with a Ka-band channel are available from several low Earth orbiting satellites that sample at different times of the day. Together they can be used to construct a diurnal cycle of brightness temperature for each location on Earth (Holmes et al., 2013b; Norouzi et al., 2012). This diurnal brightness temperature can then be scaled to match the diurnal temperature cycle as measured by TIR imagers (Holmes et al., 2015, 2016).

The methodology developed in Holmes et al (2015) was applied to create an 11-year record of MW-based LST (MW-LST) from various Ka-band sensors (See Section 2). Because this new dataset specifically includes diurnal information, it presents an opportunity to evaluate constellation-based MW-LST in a TSEB framework for estimating ET. For this purpose, we substituted MW-LST for MODIS LST in the global implementation of ALEXI as described in Hain and Anderson (in review) and generated a data record of weekly ET for the time-period 2003 to 2013. No re-calibration of ALEXI was applied in this experiment to accommodate MW-LST. The only difference between the two resulting multi-year records of ET estimates are the spectral window (MW Ka-band Vs TIR) and spatial resolution of the LST inputs (0.25° for the MW implementation: ALEXI-MW, and 0.05° for the MODIS implementation: ALEXI-IR). In order to make the subsequent comparison with ALEXI-IR as direct as possible, the MODIS





cloud mask was also applied to MW-LST. This assures that potential issues related to the applicability of the ALEXI framework during cloudy conditions (particularly its assumptions regarding boundary layer development) are separated from the question of MW-LST performance within the two-source framework. The results are discussed by region and season, and in terms of bulk ET and its inter-annual variation. With this analysis, we hope to establish the degree to which ALEXI-MW resembles the ALEXI-IR in

clear sky situations. The performance of the ALEXI model with all-sky LST observations will be the topic of subsequent investigations.

## 2 Methodology

### 2.1 ALEXI model

The ALEXI method is a comprehensive set of algorithms to diagnose the surface energy balance with the aim of retrieving ET
(Anderson et al., 2007b; Mecikalski et al., 1999). It is based on the TSEB approach (Kustas and Norman, 1999; Norman et al., 1995) in which the partitioning of turbulent fluxes is evaluated for the soil (s) and the canopy (c) separately. This is accomplished by 1) parameterizing the divergence of net radiation (*Rnet*) between canopy and soil surface and 2) attributing the observed composite surface radiometric temperature (*Trad*) into soil and canopy temperatures, $T_s$ and $T_c$ based on vegetation cover fraction. An initial guess for the canopy transpiration is based on the assumption that the green part of the canopy transpires at its potential
rate ($LE_c = LE_{PT}$), where $LE_{PT}$ is estimated with a modified Priestley and Taylor approximation (1972). The sensible heat flux for the two source components ($H_s$ and $H_c$) is then calculated in a set of equations that accounts for their different resistance to heat transfer and that satisfy the observation-based $T_s$ and $T_c$ and air temperature $T_a$ (Norman et al., 1995). The final estimate of ET is determined in an iterative procedure in which $LE_c$ is reduced until a solution is found where the soil evaporation ($LE_s$) is non-negative.

ALEXI couples TSEB with an atmospheric boundary layer model to relate the morning rise in *Trad* to the growth of the overlying planetary boundary layer and simulate an internally consistent $T_a$. This removes the need for $T_a$ as an input dataset and limits the sensitivity of the method to biases in instantaneous satellite-based temperature estimates, while allowing for regional and global implementations of the model (Anderson et al., 1997). The ALEXI model is intended for large spatial grids (~5 km grids) and provides the physical foundation to the ALEXI/DisALEXI modelling system that has been applied to many satellite-based TIR data
streams from 30-m to 10-km spatial resolutions (Anderson et al., 2011).

The experiment described in this paper is based on a recent global implementation of the ALEXI model (Hain and Anderson, in review). The data sources for this version of ALEXI are listed in Table 1. The primary input to ALEXI is $T_{rad}$ at two times during the morning: 1.5 hours after sunrise (time 1) and 1.5 hours before solar noon (time 2). This paper compares two sets of ALEXI ET estimates based on the exact same model formulation but with alternative LST inputs to estimate the time integrated change in mid-
morning $T_{rad}$. The baseline is MODIS-LST from the Moderate Resolution Imaging Spectroradiometer (MODIS) on the polar orbiting satellites Aqua and Terra from NASA's Earth Observing System (EOS) program, which is used as the input in the current global ALEXI implementation (Hain and Anderson, in review) described in Section 2.2. The alternative LST input from MW data is described in Section 2.3. The two separate implementations of ALEXI are identified by their temperature input source: ALEXI-IR (with MODIS-LST) and ALEXI-MW (with MW-LST), all other inputs needed to run ALEXI are identical for both implementations.
There are two pathways through which $T_{rad}$ affects ALEXI ET estimates: directly through the estimation of the morning rise in temperature, $\Delta T_{rad}$, which affects the boundary layer growth and the strength of the sensible heat fluxes; and indirectly through the calculation of the longwave heat loss, both at the canopy and at the soil surface, which factors in *Rnet*. Whereas the former is not sensitive to biases in mean daily temperature, the latter has a weak sensitivity to the absolute temperature at time 1 and time 2.



### 2.2 Temperature from MODIS

The MODIS instrument on the polar-orbiting Aqua satellite (July 2002 to present) with an equator overpass time of 1:30 a.m. / p.m. provides global TIR observations with spectral bands suitable for estimating LST. The specific LST product used for the ALEXI implementation is the MODIS Climate Modelling Grid (CMG) 0.05° daily LST product (MYD11C1 (Wan, 2008)), which is

distributed by the Land Processes Distributed Active Archive Center (https://lpdaac.usgs.gov). Although the overpass times of this satellite do not correspond directly with ALEXI's time 1 and time 2, Hain et al. (2017) show that GOES-based $\Delta T_{rad}$ can be estimated with a 5-10 % relative error using a tree-based regression model based on independent variables including vegetation index, and landcover class.

### 2.3 Temperature from a constellation of MW satellites

The MW-LST product is based on vertical polarized Ka-band (36-37 GHz) brightness temperature ($T^{Ka}$), a spectral band commonly included in multi-frequency microwave radiometers in low-Earth orbit. The current MW-LST product integrates observations from six of these satellites. Most important are the Advanced Microwave Scanning Radiometer EOS (AMSR-E) on Aqua from mid-2002 to October 2011 and its successor AMSR2 on the Global Change Observation Mission 1st Water (GCOM-W) from July 2012 onward. Also included are the Special Sensor Microwave and Imager (SSM/I) on platforms F13, F14 and F15 of the Defense

Meteorological Satellite Program; the Tropical Rainfall Measurement Mission (TRMM) Microwave Imager (TMI); and Coriolis-WindSat. Together this constellation of Ka-band radiometers allows for the estimation of the diurnal temperature cycle in a process that can be summarized in 4-steps, detailed below and diagrammed in Fig. 1.

### 2.3.1 Inter-calibration of MW satellites

All available Ka-band observations are combined to create a global record with up to 8 observations per day for each 0.25°

resolution grid box. The data are binned in 15 minute windows of local solar time (0:00-0:15 is first window of the day). The brightness temperatures are inter-calibrated using observations from the TRMM satellite (with an equatorial overpass) as a transfer reference. Individual 0.25° averages are masked if the spatial standard deviation of the oversampled Ka-band observations exceeds a prior determined threshold for each grid box. Both the inter-calibration and quality control procedures are described in detail in Holmes et al (2013a). The resulting global record of inter-calibrated Ka-band brightness temperatures spans the years 2003-2013.

### 2.3.2 Fitting of diurnal cycle model to sparse observations

For days with suitable observations (a minimum of 4, at least one of which is close to solar noon) and no $T^{Ka} < 250\ K$ (an indication of frozen soil), a continuous diurnal temperature cycle (DTC) is fitted. The DTC model is based on Göttsche and Olesen (2001) with slight adaptations to limit the number of parameters. This implementation (DTC3) is fully described in Holmes et al. (2015). DTC3 summarizes the DTC with four parameters: daily minimum ($T_0$) at start and end of day, diurnal amplitude $A$, and

diurnal timing $\varphi$. The fitting procedure first determines $\varphi$ as a temporal constant (Holmes et al., 2013b) and subsequently $T_0$ and $A$ for each day individually. The success of the fit ($\varepsilon_d$) is expressed as the root mean square error (RMSE) between the modelled and observed $T^{Ka}$ for the $n$ observations at times $t$ in any given day ($d$), calculated following Eq. (1):

$$\varepsilon_d = \sqrt{\frac{1}{n} \sum_{i=1}^{n} \left(T_i - DTC3(\varphi, T_0, A, t_i)\right)^2} \qquad (1)$$

This method was applied to the entire record of inter-calibrated Ka-band brightness temperatures (section 2.3.1) to create a database

of annual maps of $\varphi^{Ka}$, and daily maps of $T_0^{Ka}$ and $A^{Ka}$.



### 2.3.3 Scaling of MW DTC parameters to match TIR-LST target

To relate the diurnal cycle in Ka-band brightness temperature to the composite radiative temperature of the land surface requires a set of DTC parameters that is equivalent to those derived from $T^{Ka}$ but derived from a TIR-LST product. In the present analysis, the TIR-LST that serves as a reference is produced at the Land Surface Analysis, Satellite Application Facility (LSA-SAF). LSA-SAF

LST is based on TIR window channels of SEVIRI (Spinning Enhanced Visible and Infra-Red Imager (SEVIRI) on board the Meteosat Second Generation (MSG) geostationary satellite. The same method for fitting a DTC model to sparse observations (section 2.3.2) was applied to the LSA-SAF LST to create a database of annual maps of $\varphi^{TIR}$ and daily maps of $T_0^{TIR}$ and $A^{TIR}$ (Holmes et al., 2015). This preparation step is diagrammed in Fig 1 as step '0'.

The Ka-band DTC parameters $(T_{0,d}^{Ka}, A_d^{Ka})$ are scaled so that the long-term mean matches that of the equivalent TIR-based

parameters $(T_{0,d}^{TIR}, A_d^{TIR})$. Because $T_0^{Ka}$ is affected by the sensing depth, the scaling is performed by using daily mean temperature as an intermediate, which is defined as $(\overline{T}^{Ka} = T_0^{Ka} + A^{Ka}/2)$ for this purpose.

$$A_d^{MW} = A_d^{Ka}/\delta \tag{2}$$

$$\overline{T}_d^{MW} = \beta_0 + \beta_1 \overline{T}_d^{Ka} \tag{3}$$

The scaled parameters are indicated with the superscript '$MW$'. The parameter $\delta$ represents the slope of the zero-order least squares

regression line for estimating the amplitude of $A_d^{Ka}$ from TIR-LST ($A_d^{TIR}$). The intercept ($\beta_0$) and slope ($\beta_1$) to correct the mean daily temperature ($\overline{T}_d^{Ka}$) for systematic differences with TIR-LST ($\overline{T}_d^{TIR}$) are determined with a constrained numerical solver, as in Holmes et al. (2015). The constraint is based on radiative transfer considerations and assures that the scaling of the mean is in agreement with the prior scaling of the amplitude (Eq. 2).

The set of time-constant scaling parameters ($\delta$, $\beta_0$ and $\beta_1$) were determined for each 0.25° grid box based on all days in the period

2007-2012 where both MW and TIR-based DTC parameters were available (generally clear sky and above freezing). The consequence of using LSA-SAF LST as the reference product is that observation-based scaling parameters are limited to the domain covered by Meteosat (Africa, Europe, Middle-East). Outside this domain, the parameters must be extrapolated. The procedure for the extrapolation is still in development, and currently entails fitting linear regressions with vegetation characteristics. Because of the limited confidence in the scaling parameters outside the MSG-domain, the analysis in this paper is focused on the Africa and Europe

domain. Some results of the global set will be presented in the comparison with flux tower observations (Section 3.4).

### 2.3.4 Constructing MW-LST

Global maps of the time-constant parameters ($\delta$, $\beta_0$ and $\beta_1$, section 2.3.3) are used to calculate the daily DTC parameters ($T_{0,d}^{MW}, A_d^{MW}$) in the scaled climatology of the TIR-LST product. This scaling (Eqs. 2 and 3) is applied to every day for which estimates of $\overline{T}^{Ka}$ and $A^{Ka}$ are available (see section 2.3.2). The methodology to scale the DTC parameters from this record of Ka-

band observations to a physical temperature range is described in more detail in Holmes et al. (2015). The scaled parameters together with $\varphi^{TIR}$ are then used to construct the MW-LST based on the same DTC3 model as used in step 2:

$$\text{MW-LST}_i = DTC3(\varphi^{TIR}, T_{0,d}^{MW}, A_d^{MW}) \tag{4}$$

The use of the DTC model allows MW-LST to be diurnally complete for days when both $T_{0,d}^{MW}$ and $A_d^{MW}$ are available. MW-LST can therefore be generated at any time increment (i). The MW-LST database used for this paper was generated at 15-minute temporal

interval. This allows $T_{rad1}$ and $T_{rad2}$ to be accurately interpolated from the database. $\varepsilon_d^{Ka}$ (Eq. 1) is used to flag days where the assumptions imposed by the shape of clear sky DTC3 are not valid or individual Ka-band observations have a large bias. In this experiment, MW-LST was only used if $\varepsilon_d^{Ka}$ is 2.5 K or lower.



### 2.4 ALEXI Model Output

The output of ALEXI that we focus on in this paper is ET, specifically the 7-day total ET in mm/week (converted from MJ/week by dividing by the heat of vaporization times the density of water, both taken as constants of 2.451 MJ/kg and 1000 kg/m3 respectively). The continuous 7-day totals are achieved by temporal gap-filling of (clear sky) ET as a fraction of clear-sky latent heat

flux to incoming solar radiation (Anderson et al., 2007a). To maximize similarity, the same MODIS cloud mask is applied to the ALEXI-MW implementation so that the mechanics of standard ALEXI can be evaluated under circumstances for which it has previously been developed and validated.

The fraction of days in a year where a clear sky MODIS-based $T_{rad1}$ and $T_{rad2}$ is available for ALEXI is below 0.3 for large parts of Europe and (sub)-tropical Africa (Fig 2a). In these areas the revisit time between observation days regularly exceeds 5 days, a

threshold for temporal downscaling given the persistence of ET fraction (Alfieri et al., 2017). On average for the non-coast pixels, there is a MW-based estimate available for 69 % of those days where there is also a (clear sky) MODIS-based $T_{rad1}$ and $T_{rad2}$. This overlap in coverage can support further calibration of MW-LST to MODIS LST. In terms of potential for additional sampling through the use of MW-based LST, Fig 2c shows that MW-based estimates for the two ALEXI times are available for 54 % of the days where no MODIS-based estimate is available. Fig 2d depicts the fraction of days where either MODIS or MW-LST can be used

to estimate the $T_{rad}$ inputs required to run ALEXI. This shows that the addition of MW-LST can bring the minimum average coverage in this domain to once every two days.

### 2.5 Flux tower observations

Many flux towers include eddy-covariance (EC) measurements of ET that are commonly used for ground truthing of remote sensing and model-based ET estimates (Baldocchi et al., 2001). Harmonized Fluxnet data are distributed in so-called synthesis datasets.

They include the original observations at a half hour observation time, and aggregate values per day, weekly and monthly. For this work, we used the synthesis 2015 TIER 1 data as accessed in July 2016 (http://fluxnet.fluxdata.org/data/fluxnet2015-dataset/) to serve as a common ground reference for the evaluation of the temporal characteristics of ALEXI-MW and ALEXI-IR. In particular, the part of the dataset of interest here are the daily aggregates of ET flux which include quality control as described in Pastorello et al. (2014).

Based on these daily data, we computed the 7-day averages matching the window length of ALEXI. If not all days within a window have valid data, that window is disregarded. Overall, eddy-covariance observations of ET were available from 68 flux towers with at least one year of observations within the time period of this study.

### 2.6 Definition of regions

Although both MW and IR sets are available globally, the main analysis of this paper is focused on the domain encompassing Africa

and Europe. This is because only in that region is the scaling of MW-LST to TIR-based LST currently supported by data (see Section 2.3.3). However, purely correlation-based comparisons are much less affected by the mean absolute value of MW-LST product. Because of the limited availability of Flux tower data, we include all available stations from across the globe which allows us to double the amount of stations available for the analysis.

Within the main focus domain of this study we further highlight 11 climate-based domain subsets (see also Fig 3, bottom-right

panel):

    A.  West-African Sahel, Arid
    B.  West-African Sahel, Semi-Arid
    C.  Guinean Coast, Dry sub-Humid
    D.  Central Africa, Humid



  E. Horn of Africa, Arid

  F. Southern Africa, semi-Arid (large bias in Fig 4)

  G. Southern Africa, Arid (large bias in Fig 4)

  H. Iberia, semi-Arid

5  I. Germany, continental Humid

  J. European Russia, continental Humid, boreal forest (large bias in Fig 4)

  K. France, Humid

These regions are selected to represent a wide variety of seasonal variation in precipitation and climate class, and are based on the work of Trambauer et al. (2014). Rather than attempting to cover the entire domain with these subsets, we selected smaller subsets in

10 order to visualise the local deviations between MW and IR products that might otherwise be averaged out. We also added regions in Europe and several regions that showed a large bias in Fig. 4.

**2.7 Metrics**

Cumulative annual and seasonal fluxes are compared in terms of their relative deviation (RD (%)), calculated following Eq. 5:

$$RD = \frac{\bar{x} - \bar{y}}{(\bar{x} + \bar{y})/2} \times 100\% \qquad (5)$$

15 where $\bar{x}$ represents the mean of the MW product and $\bar{y}$ the mean of the IR product, both sampled at the same times. This relative comparison is useful because neither product represents the truth and this formulation places the deviations in context of the size of the fluxes. Still, if the ET is very small (average ET below 14 mm/month) than the denominator becomes too small and the RD is not reported. The temporal agreement between the anomalies in the IR and MW-based ET products is analysed in terms of the Pearson's correlation ($\rho$), and the spatial agreement in terms of correlation coefficient ($R^2$).

20 The temporal agreement of the weekly ET estimates is further compared relative to the flux tower observations that serve as a common reference. For this assessment, MW- and IR-based ET estimates are again compared in terms of $\rho$ but also in terms of root mean square error (RMSE) to quantify the absolute error. The RMSE is calculated following Eq. 7:

$$RMSE = \sqrt{\frac{1}{N}\sum(x - y)^2} \qquad (7)$$

where x is the satellite estimate of ET and y is the tower-based measurement of ET. N is the number of data pairs.

25 **3 Results**

**3.1 Comparing Multi-year means**

The mean average $\Delta T_{rad}$ as calculated from MW-LST deviates from that calculated from MODIS LST by 0-20 %, which leads to a spatial $R^2$ of 0.90 (Fig 4. top row). These spatial variations in mean values arise from the different calibration targets. MW-LST is calibrated to match the LSA-SAF LST from MSG (Europe and Africa) with a precision of 2-3K (see Section 2.3.3), and MODIS

30 $\Delta T_{rad}$ is trained on GOES (North-America) with an estimated precision of 5-10 % (see Section 2.2). These different calibration domains together with likely calibration differences between GOES and MSG LST products present sources of bias that can explain the regional variation we see in Fig 4. For example, the difference between $\Delta T_{rad}$ estimates in the North-East corner of this map may be an artefact of scaling with high incidence angles ($\theta$) for the MSG geostationary satellite. In the farthest corner ($\theta > 60°$), MSG observations were not used and the MW scaling is extrapolated based on land surface characteristics. The MW $\Delta T_{rad}$ also exceeds

35 IR-based estimates by more than 10 % in Southern Africa, for which we do not currently have an explanation.

The general agreement in mean $\Delta T_{rad}$ translates into a high agreement between IR and MW-based ALEXI in terms of mean annual ET for the period 2003 – 2011. The spatial correlation between MW and IR in terms of ET is 92 % (Fig 4. bottom row). The slightly higher agreement is likely due to physical constraints imposed by the ALEXI retrieval model (e.g. 0<LE<LE$_{PT}$). Regions with





notable differences are Boreal Russia, where MW is lower by ~ 20 %, and the Horn of Africa where MW is higher by 20-30%. MW ET is also much lower than IR ET in the Alps. Some, but not all of these features are explained by matching opposite bias between IR and MW-based $\Delta T_{rad}$.

### 3.2 Regional/Seasonal Bulk Flux comparison

Figure 5 provides a more detailed comparison between the MW and IR products for the domain subsets as described in Section 2.6. For each domain subset, it shows the mean monthly total ET and the associated monthly means of $\Delta T_{rad}$. For European Russia (region J) and to a lesser extend Germany (I) and France (K), this shows higher MW-based $\Delta T_{rad}$ resulting in lower summertime ET estimates than for ALEXI-IR. Conversely, in the wintertime the lower MW-based $\Delta T_{rad}$ results in higher ET estimates compared to the TIR products in October-December. For Iberia (H), the semi-Arid Sahel (B) and Spain (H) there appears to be a difference in

timing with MW estimating a later time for peak ET. The higher MW ET estimates in the Horn of Africa are rather uniform over the year, except for December and January where the difference is small. The size of the bias in ET for the Horn of Africa is surprising compared to the relatively modest bias in $\Delta T_{rad}$. Another disconnect between $\Delta T_{rad}$ deviation and ET can be seen in Southern Africa (regions F and G). Despite a general overestimation of $\Delta T_{rad}$ by MW the ET estimates are very close to those of ALEXI-IR. The small difference in ET estimates are the opposite of what would be expected from the $\Delta T_{rad}$ deviation. Finally, the humid

tropical climates of Guinean coast and central Africa (regions C and D) have very little differences in both $\Delta T_{rad}$ and ET.
For spatial context to these observations, the three-month total ET (averaged over 2003-2011) is calculated for December-January-February (DJF), March-April-May (MAM), June-July-August (JJA) and September-October-November (SON), see Fig 6. This shows that the cold season overestimation of MW-based ET, seen in the European regions, is present not only in Europe but also in East and Southern Africa in SON. The underestimation of MW-based ET in summer is not as pronounced in terms of its relative

difference. The apparent difference in timing, seen in the Sahel and Iberian regions, shows up across the southern border of the Sahara – MW-ET is higher in MAM, and TIR-ET is higher in JJA. The spatial correlation between MW and IR is higher in SON (96 %) and DJF (97 %) compared to the periods MAM (83 %) and JJA (84 %).

### 3.3 Inter-annual Variation

  Because the long term mean of MW-LST is calibrated to match a TIR reference (see Section 2.3.3), the comparison in terms of

anomalies is the real test of its performance in the ALEXI framework, especially in areas that are water limited (see Fig. 7). Of the subsets in water-limited regions, the Horn of Africa (ρ=0.78) and Spain (ρ=0.85) subsets show a high degree of correlation between MW and TIR-based ET anomalies. Semi-Arid Southern Africa (F) and the Sahel (B) show relatively poor correlation with ρ=0.48 and ρ=0.63 respectively. The size of the anomaly is much larger for ALEXI-MW in Southern Africa in January and February, reflecting a much larger inter-annual variation.

In energy-limited areas when ET is fully determined based on the meteorological forcing data, the effect of LST inputs is minimal. This is apparent in the Tropical region, where MW and ALEXI-IR have a correlation of 0.99 in Central Africa (region D). Figure 8 shows a map of the correlation between 3-month anomalies of MW and IR-based ALEXI ET.
Seasonal anomalies are calculated by taking the seasonal total ET for a given year and subtracting its corresponding long-term mean seasonal total (2003-2011 period, as shown in Fig 6). Examples of this are shown for a dry year (2008) and a wet year (2011), see

Fig 9. Overall the two sets of anomalies agree very well – the MW ALEXI appears to identify roughly the same areas with anomalous high or low ET. The agreement is better in the wet year than in the dry year.





### 3.4 Comparison with flux tower observations

The availability of eddy-covariance observations of ET from 68 flux towers allows for a more detailed grid-level analysis of temporal agreement. The metrics ρ and RMSE were computed for each flux tower site and listed in Table 2. Overall the mean ρ is very similar for IR and MW ALEXI (ρ=0.76/RMSE=24 Vs ρ=0.74/RMSE=24). At individual stations, the differences may be more

pronounced. For ALEXI-IR, the majority of the sites have ρ between 0.6 and 0.92, and RMSE of 12-33 mm/week. The metrics for ALEXI-MW are very similar, see also Fig 10. Surprisingly, at many sites the 0.25° resolution MW product actually outperforms the 0.05° IR product in terms of ρ. Nevertheless, the ρ is generally a bit higher for ALEXI-IR, even though the average RMSE comes out the same (24 mm/week).

Some of the higher values of ρ seen for ALEXI-MW might counterintuitively be explained by its coarser resolution. When 0.05°

ALEXI-IR is also averaged over its surrounding 0.25° grid (the average of the 5x5 0.05° grid cells) there is an overall improvement in ρ (and not in RMSE). Only at three sites does this spatial degradation lower the ρ between the site and the 0.05° grid average higher than with the 0.25° grid box. For most stations the spatial degradation actually improves the ρ with the site. In fact, 40 % of the difference in ρ between MW and IR ALEXI is explained by the change in ρ from ALEXI-IR (at 0.05°) to ALEXI-IR (at 0.25°). This indicates the presence of spatially uncorrelated noise in the 0.05° MODIS that negates any positive effect of its resolution

advantage compared to a 0.25° grid average for most sites.

It is interesting to investigate what drives the difference in temporal correlation at individual sites. The second row in Fig. 10 shows how the same data as presented in Fig. 10, but broken out based on geographic domain, climate or spatial agreement. The first panel splits the sites by geographic region. Europe and Africa (MSG domain, blue) is where MW-LST was calibrated, but the relative improvement in ρ is higher in the North-American sites (GOES domain, green). This is despite the MODIS ALEXI-IR being

calibrated with GOES data. Panel 2 separates the sites based on climate, particularly in terms of the potential ET (PET) relative to the annual precipitation (P). The PET is calculated following Priestley-Taylor (1972) with an alpha is 1.26 and zero ground heat flux. Sites with humid climates (energy limited: PET<=P) have generally a higher ρ and no overall difference in ρ with station data. In arid climates (water limited: PET>P) the correlations between satellite estimate and tower observation are much lower and the ALEXI-IR has an advantage over ALEXI-MW. Further subdividing the sites in arid locations based on information on spatial

heterogeneity reveals a larger separation of performance (Fig 10, Panel 3 on bottom row). Taking the absolute bias (|b|) between ALEXI-IR at the 0.05° grid cell encompassing the tower site and the mean of the 0.25° surrounding grid box as proxy for spatial heterogeneity, we can see that for the sites that are both in a water limited region and have a high spatial bias, 11 in total, the average ρ for ALEXI-MW (ρ=0.55) is markedly lower than that for ALEXI-IR (ρ=0.65).

Six of the 68 sites have a markedly higher ρ with ALEXI-IR than with ALEXI-MW. All but one of these sites have an arid climate

(See Table 2), and four of those stations also have a high spatial bias between the 0.05° grid box and 0.25° grid box mean (|b|>2 mm/week):

- US-Ton and US-Var (PET/P=2.5/1.7, b=-6.4), Woody Savannas, same 0.05 box. The MODIS has a ρ=0.80/0.5, while the MW has a poor 0.56/0.09. Especially at US-Var, the site has an abrupt collapse of ET at end of summer that. The satellite data misses this, especially the ALEXI-MW(0.25°) product.

- Zambia, Savannas. ZM-Mon. Water limited (PET/P=2.3), spatial heterogeneous (b=-2.9)
- ES-LgS, Woody Savannas (b=11, PET/P=2.8). The MODIS has a high ρ=0.84, while the MW has a poor ρ=0.62. The average comes in at ρ=0.82. This site is located on a mountain ridge. The smaller grid size of ALEXI-IR is able to capture the vegetation conditions at the mountain ridges whereas the 0.25 grid of ALEXI-MW has more bare soil which leads to lower ET values.

The station in Sudan (SD-Dem) is the only of these 6 stations that is in a water limited region (arid desert climate) and has low spatial bias. Despite the low bias, the station ET estimates are 2.5 times satellite estimates, so it could be that the near station land use is not representative of the wider area.





The final station that shows a large advantage in ρ for ALEXI-IR relative to ALEXI-MW is Fi-Hyy (No. 63 in Table 2) in a cold region climate. It is also one of only two stations with data availability at high latitude (above 60°N). This station has land cover dominated by evergreen needleleaf forest. The bias between the 0.05° and 0.25° grid box mean is also small (b=-0.6). The MW observations have relatively many weeks with very low ET estimates compared to the ALEXI-IR. The reason for this is not readily apparent but it could be that the MW product suffers from rainclouds that suppress temperature estimates during the morning hours around ALEXI time 1. This, in turn, leads to an overestimated morning temperature rise.

In contrast to these sites, there are two sites where the ALEXI-MW outperforms ALEXI-IR in terms of correlation with in situ sites despite being in a relatively arid climate with large spatial bias: US-SRG, US-NR1. For US-NR1, ρ is low because station records high values in winter time, and the site is located in an evergreen forest east of a mountain ridge, with high day to day variation, possibly due to varying wind direction or shading effects. Despite this, both satellite products pick up the seasonal cycle reasonably well, except that they both underestimate wintertime ET.

### 3.5 Prospects for merging MW- and TIR-based ALEXI

The above analysis of flux tower data suggests that the prospect for merging MW and TIR is good for humid climate regions and also for more arid regions but with the additional condition of requiring a more spatially homogeneous landscape at the 0.25° grid scale. In arid climates, the overall correlation between satellite retrievals and tower sites is relative poor. This is expected because the signal to noise is reduced in areas with low seasonal variation in ET. The change in input LST from IR to MW reduces the ρ on average by 0.04 in the arid climates. Even so, it seems that only if there is also local heterogeneity at the 0.25° grid resolution will the prospect for merging ALEXI-MW and ALEXI-IR require more complex procedures to avoid degradation relative to the ALEXI-IR generate at 0.05° grid resolution.

In fact, as a simple test of a potential merging of MW and IR-based ALEXI ET estimates we calculated the simple average of MW (at 0.25° resolution) and TIR-based (at 0.05° resolution) 7-day ALEXI ET totals (ALEXI-IR+MW) and re-calculated the correlation and RMSE with the flux tower sites. Fig 12 shows a visual comparison of these temporal metrics with the tower observations, analogous to Fig. 10 but now with the results for ALEXI-IR+MW on the y-axis instead of ALEXI-MW independently. In the top, left-hand panel, most stations now appear below the 1:1 line, showing a general increase in ρ as a result of simply averaging ALEXI-IR and ALEXI-MW. Indeed, by averaging the weekly ET estimates the ρ is improved from an average of 0.76 for ALEXI-IR to 0.78 for ALEXI-IR+MW. The change in correlation is significant for 12 of the 68 stations (according to a Fischer z-test), and those stations improve from an average of 0.69 for ALEXI-IR to 0.75 for ALEXI-IR+MW (see also Table 2). The top right-hand panel shows a comparison of RMSE values calculated between ALEXI-IR and the station data, and between ALEXI-IR+MW and the station data. Values above the line indicate a reduction in RMSE for the combined product compared to ALEXI-IR product. The individual station results are all close to the 1:1 line, and the mean RMSE reduces from 24 mm/week to 23 mm/week. The bottom row of the figure shows results broken down by subsets of stations based on domain, climate or spatial bias as was done for Fig 10, above. This shows when taking the simple average of ALEXI-IR and ALEXI-MW, the overall correlation improves for all but the most challenging conditions. Even for the subset where the relative performance for ALEXI-MW was poorest (stations water limited climate and with a spatial bias between the 0.05° and 0.25° grid box), the correlation is very close to that of the ALEXI-IR alone (ρ=0.64 vs. ρ=0.65).

### 4 Discussion and Conclusion

This paper shows that a newly developed MW-LST product can be used to effectively substitute TIR-based LST in a two-source energy balance approach to estimate coarse-resolution ET (~25 km) from space. This particular TSEB approach, the ALEXI model



framework, is an approach that minimizes sensitivity to absolute biases in input records of LST through the analysis of the rate of change in morning LST. It is therefore an important test of the ability to retrieve diurnal temperature information from a constellation of satellites that provide 6-8 observations of Ka-band brightness temperature per location per day. This represents the first ever attempt at a global implementation of ALEXI with MW-based LST and is intended as the first step towards providing all-

weather capability to the ALEXI framework.

Because the long-term diurnal features of MW-LST are calibrated to TIR-LST, it is perhaps not surprising that the long-term bulk ET estimates agree with a spatial correlation of 92 % for total ET in the Europe/Africa domain. A comparison with biases in the input datasets of $\Delta T_{rad}$ shows that a large part of the remaining differences can be mitigated by specifically calibrating MW-LST to MODIS LST. More convincing is the agreement in seasonal (3-month) averages of and 83-97 % because the calibration is based on

time-constant parameters. Adding another layer of challenging complexity is the comparison in terms of 3-month anomalies. By this test, ALEXI-MW also matches ALEXI-IR very closely, demonstrating an ability to capture the development and extent of drought conditions.

The two parallel ALEXI implementations are further compared at the maximum temporal resolution of the current ALEXI output (7 days) and relative to a common ground measured reference provided by the FLUXNET consortium. The 68 stations that were

available for this analysis represent a wide range of land cover characteristics and climate conditions. Overall, they indicate a surprisingly close match in both performance metrics (ρ and RMSE) for all but the most challenging sites in terms of spatial heterogeneity and level aridity. Spatial heterogeneity places an obvious penalty on ALEXI-MW due to the coarser MW-LST input, even though in general ALEXI-IR improves in terms of its correlation with the tower data when it is spatially downgraded to 0.25° resolution. In terms of climate, the sensitivity of ALEXI ET retrievals to the LST input increases with increasing water limitation of

evapotranspiration. In humid climates, the ET estimate is close to the potential ET, which is calculated based on meteorological forcing.

As an experiment, we also calculated the average of ALEXI-IR and ALEXI-MW, both at their original spatial resolutions (0.05° and 0.25° respectively) and calculated the same temporal performance metrics with the station observations. Since they are derived from completely difference remote sensing techniques, errors in the temporal features of MW-LST should be independent of those in TIR-

LST. Moreover, because LST is one of the main diagnostic inputs to ALEXI the LST related errors in retrieved ET are also expected to be independent. This, in turn, is expected to allow for a reduced estimation uncertainty of ET by averaging two independent estimates of the same variable. The overall observed improvement from ρ=0.76 to ρ=0.78 confirms this expectation that part of the error in ET estimates from ALEXI-IR and ALEXI-MW is independent.

Based on the analyses presented in this paper, we outline the following roadmap for an all-sky implementation of ALEXI-MW. First

of all, there is a need for global observation based calibration of MW-LST with MODIS-LST to reduce biases as identified at the high incidence angles of the MSG domain and avoid the need for extrapolation of scaling parameters. Second, the MW-LST could be used to improve the TIR cloud mask by attributing anomalous TIR-based $\Delta T_{rad}$ to the presence of clouds, with subsequent improvements in ALEXI-IR ET estimates. Finally, it appears that a simple averaging of ALEXI-IR and ALEXI-MW would provide for a reduction in estimation uncertainty of ALEXI ET for times when both are available. With a combined MW+IR ALEXI

estimates it appears entirely feasible to reduce the current window length for reporting MODIS ALEXI ET totals from 7-days to as low as 2. At a window length of 2 days the average satellite coverage would support each 2-day total with at least one observation (See Fig. 2). This would reduce the reliance on temporal downscaling and its associated assumptions and impact on estimation error. More independent estimates of ET would allow for more robust statistical analysis in the context of land-atmosphere exchange studies, even if the record length is not extended. Perhaps most importantly, a shorter reporting interval would also allow for earlier

detection of agricultural drought as reflected in the ET-based drought indices (Anderson et al., 2011).



## Data Availability

The ALEXI-IR data is available from NASA SPoRT (MSFC). The ALEXI-MW is an intermediate research product available upon request. Time-series of ALEXI-MW and ALEXI-IR covering the site locations and time period of this paper are available upon request from the corresponding author. The Flux tower data is publicly available through the FLUXNET community as detailed in Section 2.4.

## Acknowledgements

This work was funded by NASA through the research grant "The Science of Terra and Aqua" (13-TERAQ13-0181).

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





**Table 1. Primary inputs for current global implementation of ALEXI**

| Data | Purpose | Source | Spatial Resolution |
|---|---|---|---|
| LST | $T_{rad}$, Net Radiation | MODIS (MYD11C1) (Section 2.2) | 0.05° |
| | | MW-LST (Section 2.1) | 0.25° |
| Surface Shortwave and Longwave Radiation Fluxes | Net Radiation | CFS-R[2], CFSv2[3] | 0.5° |
| Albedo | Net Radiation | MODIS (MOD43C)[5] | 0.05° |
| LAI | Trad partitioning | MODIS (MOD15A)[6] | 0.01° |
| Landcover type | Canopy characteristics | UMD[7] | 0.01° |
| Wind speed | Aerodynamic resistance | CFS-R | 0.5° |
| Lapse rate profile | Atmospheric Boundary layer growth model | CFS-R | 0.5° |

[2]NCEP Climate Forecast System Reanalysis (Saha and et al, 2010), [3] (Saha and al, 2011), [4] (Doelling, 2012), [5](Moody et al., 2005), [6](Myneni et al., 2002), [7](Hansen et al., 2000)



**Table 2. Time-series correlation of flux tower ET observations with three alternative satellite-based ET estimates (ALEXI_IR, ALEXI-MW, ALEXI- IR+MW). Comparison is based on weekly averages in the period of 2003 to 2011 (number of data pairs is noted in the table). Bias is mean difference between 0.05 grid and 0.25 surrounding grid average as measured by ALEXI-IR.**

| No. | Site ID | Latitude | Longitude | IGBP Vegetation Type | Climate PET/P | bias | Data pairs (wk) | ALEXI-IR (0.05) R | RMS | ALEXI-MW (0.25) R | RMS | ALEXI-IR+MW (Avg) R | RMS | Fischer Z-test |
|---|---|---|---|---|---|---|---|---|---|---|---|---|---|---|
| 1 | IT-PT1 | 45.20087222 | 9.061038889 | Croplands | 1.09 | 1.63 | 78 | 0.94 | 16.75 | 0.92 | 19.31 | 0.94 | 17.78 | 0 |
| 2 | CZ-wet | 49.02465 | 14.77035 | Croplands | 0.94 | 1.65 | 101 | 0.92 | 12.97 | 0.88 | 15.84 | 0.91 | 13.68 | 0 |
| 3 | US-MMS | 39.32315 | -86.413139 | DBF | 0.68 | 3.91 | 439 | 0.91 | 15.89 | 0.88 | 18.64 | 0.91 | 16.34 | 0 |
| 4 | DE-Hai | 51.07916667 | 10.453 | Mix. Forest | 0.68 | 3.15 | 322 | 0.91 | 13.00 | 0.87 | 19.75 | 0.91 | 14.55 | 0 |
| 5 | CH-Fru | 47.11583333 | 8.537777778 | Mosaic | 0.39 | 3.77 | 241 | 0.90 | 21.64 | 0.80 | 28.58 | 0.89 | 23.43 | 0 |
| 6 | US-Syv | 46.242017 | -89.34765 | Mix. Forest | 0.87 | 0.57 | 171 | 0.88 | 17.85 | 0.91 | 15.58 | 0.91 | 16.23 | 0 |
| 7 | DE-Gri | 50.94946944 | 13.512525 | Mix. Forest | 0.52 | -0.38 | 329 | 0.88 | 12.22 | 0.88 | 12.29 | 0.89 | 11.59 | 0 |
| 8 | US-Ne1 | 41.165056 | -96.476638 | Croplands | 0.89 | -0.86 | 427 | 0.88 | 33.98 | 0.87 | 31.74 | 0.89 | 32.55 | 0 |
| 9 | US-Ne3 | 41.179667 | -96.439646 | Croplands | 1.00 | -0.42 | 420 | 0.88 | 27.37 | 0.87 | 25.85 | 0.89 | 26.24 | 0 |
| 10 | DE-Obe | 50.7836167 | 13.7196306 | ENF | 0.53 | 3.09 | 167 | 0.87 | 13.41 | 0.83 | 14.03 | 0.86 | 13.17 | 0 |
| 11 | CH-Cha | 47.21022222 | 8.410444444 | Mosaic | 0.46 | -5.44 | 257 | 0.87 | 38.76 | 0.89 | 29.28 | 0.92 | 32.94 | 1 |
| 12 | AU-Ade | -13.0769 | 131.1178 | Savannas | N/A | -1.12 | 77 | 0.87 | 50.98 | 0.84 | 43.32 | 0.88 | 46.69 | 0 |
| 13 | US-Ne2 | 41.164871 | -96.4701 | Croplands | 0.87 | -0.86 | 434 | 0.87 | 32.12 | 0.85 | 30.41 | 0.87 | 30.93 | 0 |
| 14 | DE-Tha | 50.96361111 | 13.56694444 | ENF | 0.65 | 1.63 | 413 | 0.86 | 14.01 | 0.88 | 11.79 | 0.88 | 12.21 | 0 |
| 15 | US-ARc | 35.54649 | -98.04006 | Grasslands | 1.27 | 1.87 | 84 | 0.86 | 37.53 | 0.89 | 35.10 | 0.89 | 36.11 | 0 |
| 16 | AU-Tum | -35.6566 | 148.1516 | EBF | 0.96 | 2.52 | 289 | 0.85 | 22.61 | 0.79 | 24.60 | 0.83 | 23.17 | 0 |
| 17 | BE-Vie | 50.30507 | 5.998052 | Mix. Forests | 0.61 | 0.47 | 274 | 0.85 | 13.20 | 0.74 | 16.23 | 0.83 | 13.30 | 0 |
| 18 | DE-Geb | 51.1001 | 10.9143 | Croplands | 0.81 | -1.18 | 411 | 0.85 | 15.53 | 0.77 | 19.36 | 0.83 | 16.70 | 0 |
| 19 | DE-Seh | 50.8706233 | 6.44965306 | Croplands | 0.63 | 0.48 | 132 | 0.84 | 24.10 | 0.80 | 29.29 | 0.85 | 26.21 | 0 |
| 20 | BE-Lon | 50.55219444 | 4.744772222 | Croplands | 0.61 | 0.09 | 309 | 0.84 | 15.38 | 0.86 | 15.30 | 0.86 | 14.89 | 0 |
| 21 | AU-Wom | -37.4222 | 144.0944 | EBF | 0.78 | 0.65 | 77 | 0.84 | 29.12 | 0.82 | 25.16 | 0.86 | 26.40 | 0 |
| 22 | ES-LgS | 37.097936 | -2.965833 | Woody Sav. | 2.01 | 11.28 | 92 | 0.84 | 11.31 | 0.62 | 18.33 | 0.82 | 12.25 | 0 |
| 23 | DE-Kli | 50.8928806 | 13.5225056 | Croplands | 0.46 | -1.26 | 284 | 0.84 | 14.29 | 0.84 | 15.43 | 0.86 | 14.08 | 0 |
| 24 | FI-Hyy | 61.8475 | 24.295 | ENF | 0.84 | -0.61 | 312 | 0.84 | 13.41 | 0.65 | 22.53 | 0.80 | 15.99 | 0 |
| 25 | DE-Lkb | 49.09961667 | 13.30466667 | ENF | 0.45 | 1.38 | 92 | 0.84 | 14.62 | 0.79 | 15.03 | 0.83 | 13.91 | 0 |
| 26 | RU-Fyo | 56.4615278 | 32.9220833 | Mix. Forests | 0.80 | 0.82 | 314 | 0.83 | 17.49 | 0.71 | 25.18 | 0.81 | 20.13 | 0 |
| 27 | US-ARb | 35.54974 | -98.04023 | Croplands | 1.28 | 1.87 | 78 | 0.82 | 31.03 | 0.89 | 28.30 | 0.88 | 29.44 | 0 |
| 28 | US-Blo | 38.89525 | -120.63275 | ENF | 0.82 | 0.81 | 206 | 0.82 | 27.89 | 0.89 | 21.47 | 0.87 | 24.11 | 0 |
| 29 | AU-DaP | -14.0633 | 131.3181 | Savannas | 0.94 | -0.85 | 190 | 0.82 | 37.68 | 0.70 | 36.96 | 0.81 | 36.29 | 0 |
| 30 | FR-Gri | 48.84422 | 1.95191 | Croplands | 0.75 | -1.18 | 232 | 0.82 | 21.57 | 0.81 | 19.83 | 0.82 | 20.40 | 0 |
| 31 | IT-MBo | 46.014678 | 11.045831 | Grasslands | 0.52 | 3.15 | 419 | 0.82 | 20.90 | 0.87 | 17.58 | 0.86 | 18.13 | 1 |
| 32 | IT-Lav | 45.9562 | 11.28132 | ENF | 0.65 | 0.99 | 395 | 0.81 | 18.75 | 0.78 | 22.63 | 0.82 | 19.62 | 0 |
| 33 | CA-SF3 | 54.09156 | -106.00526 | ENF | 0.91 | 2.21 | 155 | 0.81 | 16.04 | 0.80 | 17.46 | 0.82 | 15.69 | 0 |
| 34 | BE-Bra | 51.30916667 | 4.520555556 | Mix. Forests | 0.57 | -0.74 | 308 | 0.81 | 13.99 | 0.73 | 14.11 | 0.80 | 12.76 | 0 |
| 35 | AU-Wac | -37.429 | 145.18725 | EBF | 0.13 | 8.53 | 132 | 0.80 | 20.11 | 0.80 | 20.61 | 0.82 | 19.64 | 0 |
| 36 | US-Ton | 38.4316 | -120.966 | Woody Sav. | 1.86 | -6.37 | 353 | 0.80 | 16.06 | 0.56 | 22.76 | 0.71 | 16.24 | 1 |
| 37 | US-WCr | 45.80592667 | -90.07985917 | DBF | 0.82 | 0.04 | 194 | 0.78 | 20.52 | 0.82 | 19.33 | 0.83 | 18.38 | 0 |
| 38 | CA-SF1 | 54.48495 | -105.81735 | ENF | 0.93 | 0.52 | 130 | 0.78 | 24.57 | 0.70 | 32.84 | 0.78 | 27.94 | 0 |
| 39 | US-Twt | 38.1055 | -121.6521 | Croplands | 1.68 | 1.89 | 108 | 0.78 | 73.06 | 0.72 | 74.07 | 0.77 | 73.45 | 0 |
| 40 | AT-Neu | 47.1166687 | 11.31750011 | Mix. Forests | 0.67 | -1.26 | 380 | 0.78 | 22.80 | 0.83 | 23.67 | 0.87 | 21.19 | 1 |
| 41 | AU-DaS | -14.159281 | 131.388 | Savannas | 0.85 | 3.09 | 178 | 0.77 | 32.28 | 0.70 | 25.55 | 0.77 | 28.23 | 0 |
| 42 | ZM-Mon | -15.43777778 | 23.25277778 | Savannas | 1.68 | -2.88 | 92 | 0.76 | 24.78 | 0.53 | 26.77 | 0.72 | 24.97 | 0 |
| 43 | SD-Dem | 13.2829 | 30.4783 | Grasslands | 3.13 | 1.19 | 112 | 0.76 | 41.89 | 0.47 | 42.99 | 0.68 | 42.16 | 0 |
| 44 | US-AR1 | 36.4267 | -99.42 | Grasslands | 1.42 | 3.06 | 131 | 0.75 | 31.03 | 0.78 | 24.87 | 0.79 | 27.20 | 0 |
| 45 | IT-Ren | 46.58686 | 11.43369 | ENF | 0.62 | 0.53 | 360 | 0.75 | 33.01 | 0.77 | 34.51 | 0.81 | 32.61 | 1 |
| 46 | US-Los | 46.08268 | -89.97919 | Mix. Forests | 0.71 | 0.06 | 271 | 0.75 | 19.32 | 0.86 | 23.71 | 0.84 | 19.89 | 1 |
| 47 | US-AR2 | 36.6358 | -99.5975 | Grasslands | 1.48 | 0.38 | 121 | 0.75 | 20.19 | 0.76 | 15.66 | 0.78 | 15.98 | 0 |
| 48 | US-Me2 | 44.4523 | -121.5574 | ENF | 2.07 | 0.35 | 374 | 0.73 | 30.89 | 0.72 | 28.37 | 0.75 | 29.19 | 0 |
| 49 | US-ARM | 36.6058 | -97.4888 | Croplands | 1.21 | -3.34 | 409 | 0.72 | 21.70 | 0.69 | 19.27 | 0.73 | 19.13 | 0 |
| 50 | IT-Col | 41.84936 | 13.58814 | DBF | 0.88 | -0.67 | 180 | 0.72 | 20.37 | 0.79 | 18.08 | 0.77 | 18.46 | 0 |
| 51 | NL-Loo | 52.166581 | 5.743556 | ENF | 0.83 | -0.70 | 380 | 0.70 | 28.93 | 0.67 | 26.11 | 0.71 | 27.14 | 0 |
| 52 | CA-SF2 | 54.25392 | -105.8775 | Mix. Forests | 1.82 | 0.77 | 114 | 0.70 | 25.46 | 0.59 | 32.00 | 0.70 | 27.52 | 0 |
| 53 | US-Whs | 31.743833 | -110.052222 | Open Shrub | 3.32 | 4.02 | 228 | 0.69 | 17.40 | 0.64 | 20.05 | 0.73 | 15.33 | 0 |
| 54 | CA-Qfo | 49.69247 | -74.34204 | ENF | 0.56 | -0.32 | 308 | 0.69 | 14.79 | 0.82 | 11.59 | 0.78 | 12.10 | 1 |
| 55 | US-SRG | 31.7894 | -110.8277 | Grasslands | N/A | 6.65 | 178 | 0.67 | 28.16 | 0.74 | 25.35 | 0.76 | 26.37 | 0 |
| 56 | US-NR1 | 40.03287778 | -105.5464028 | ENF | 1.11 | -4.20 | 449 | 0.64 | 20.97 | 0.73 | 24.99 | 0.73 | 20.87 | 1 |
| 57 | ZA-Kru | -25.0197 | 31.4969 | Savannas | 2.85 | -2.33 | 180 | 0.62 | 30.34 | 0.57 | 31.12 | 0.65 | 30.06 | 0 |
| 58 | IT-Tor | 45.84444 | 7.578055 | ENF | 0.72 | -2.27 | 122 | 0.62 | 31.77 | 0.61 | 40.20 | 0.69 | 34.60 | 0 |
| 59 | CN-Du2 | 42.04606667 | 116.2836111 | Grasslands | 1.99 | 1.10 | 101 | 0.62 | 24.80 | 0.68 | 24.83 | 0.68 | 24.45 | 0 |
| 60 | US-Wkg | 31.736527 | -109.94188 | Grasslands | 2.78 | 0.67 | 389 | 0.61 | 19.02 | 0.73 | 15.46 | 0.74 | 16.72 | 1 |
| 61 | AU-Stp | -17.1508 | 133.3503 | Grasslands | 1.54 | -1.02 | 142 | 0.59 | 40.09 | 0.72 | 31.99 | 0.74 | 35.40 | 1 |
| 62 | AU-Dry | -15.2588 | 132.3706 | Savannas | 6.63 | 4.41 | 107 | 0.56 | 32.19 | 0.48 | 26.60 | 0.56 | 28.61 | 0 |
| 63 | CH-Dav | 46.81533333 | 9.855916667 | ENF | 0.75 | -0.14 | 413 | 0.54 | 26.50 | 0.59 | 29.89 | 0.61 | 26.39 | 0 |
| 64 | US-Var | 38.40666667 | -120.9507333 | Woody Sav. | 1.28 | -6.37 | 428 | 0.50 | 21.05 | 0.09 | 34.23 | 0.29 | 25.92 | 1 |
| 65 | US-SRM | 31.82143 | -110.86611 | Open Shrub | 2.78 | -1.89 | 397 | 0.50 | 28.27 | 0.67 | 19.58 | 0.68 | 23.05 | 1 |
| 66 | FI-Sod | 67.36186111 | 26.63783333 | ENF | 0.49 | -0.08 | 185 | 0.47 | 22.50 | 0.43 | 25.90 | 0.47 | 23.69 | 0 |
| 67 | US-GLE | 41.3644 | -106.2394 | ENF | 0.41 | 10.44 | 261 | 0.46 | 27.15 | 0.37 | 36.14 | 0.46 | 28.85 | 0 |
| 68 | US-Cop | 38.09 | -109.39 | Grasslands | 4.33 | -2.16 | 87 | 0.22 | 11.83 | 0.33 | 9.82 | 0.37 | 8.78 | 0 |

ENF: Evergreen Needleleaf Forest, DBF: Deciduous Broadleaf Forest, EBF: Evergreen Broadleaf Forest, Mosaic: Cropland/Natural Vegetation Mosaic





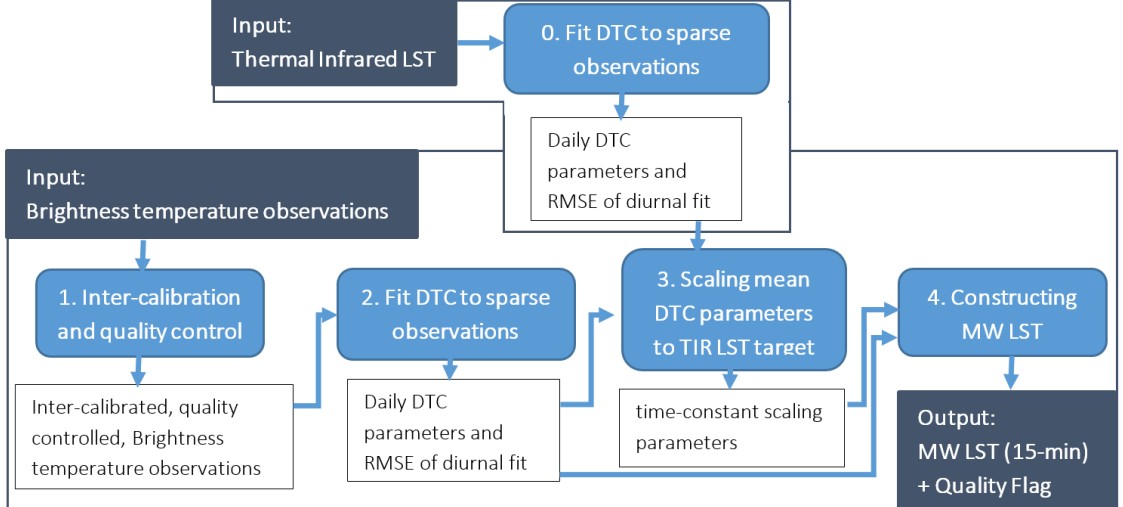

**Figure 1: MW-LST workflow**

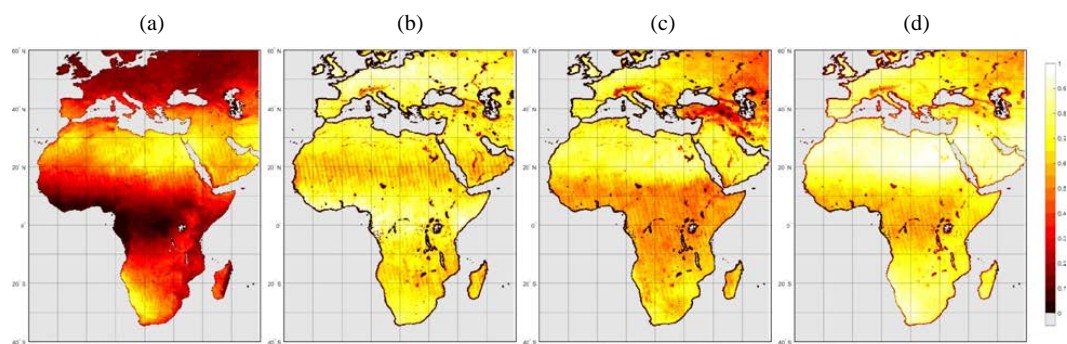

**Figure 2: Temporal coverage of MW and IR-based $\Delta T_{rad}$ in 2004. Panel a shows the fraction of total days where MODIS-based estimates of $\Delta T_{rad}$ are available. Panel b shows the fraction of this subset of days where there is also a MW-based $\Delta T_{rad}$ available. Panel c shows the fraction of days without a MODIS-based estimate but with availability of a MW-based estimate (potential for IR-gap coverage). Panel d shows the fraction of total days where either a MODIS- or a MW-based $\Delta T_{rad}$ is available.**





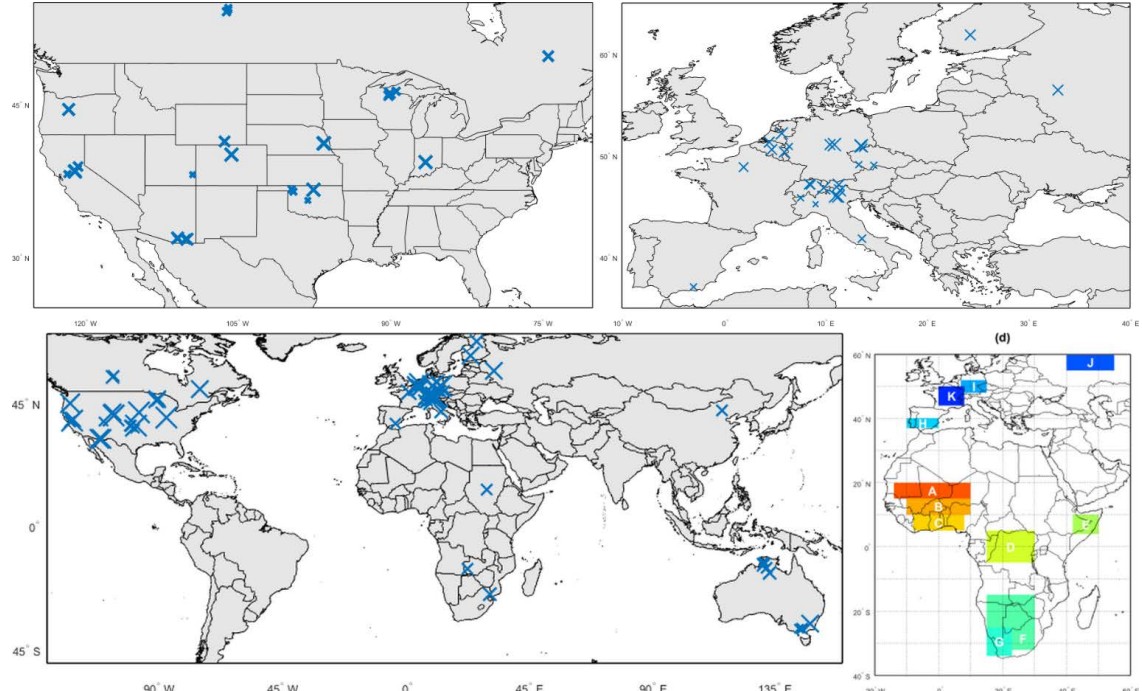

**Figure 3: Location of flux tower sites used in the analysis (see also Table 2): (a) North America, (b) Europe, (c) World. The size of the marker is in proportion to number of data days used in assessment. Panel (d) indicates the 12 regions selected based on annual precipitation cycle and geographic diversity.**







**Figure 4: Multi-year mean of $\Delta T_{rad}$ (top row, 2004-2011) and mean annual ET (bottom row, 2003-2011) for IR and MW. The transect shows the latitudinal average for longitude 10°W to 35°E. The right-hand panel shows the corresponding relative difference (RD) between the two estimates (MW - IR). Note the reversed colour bars for $\Delta T_{rad}$ and ET to emphasize their negative correlation ($\Delta T_{rad}$ up, ET down).**





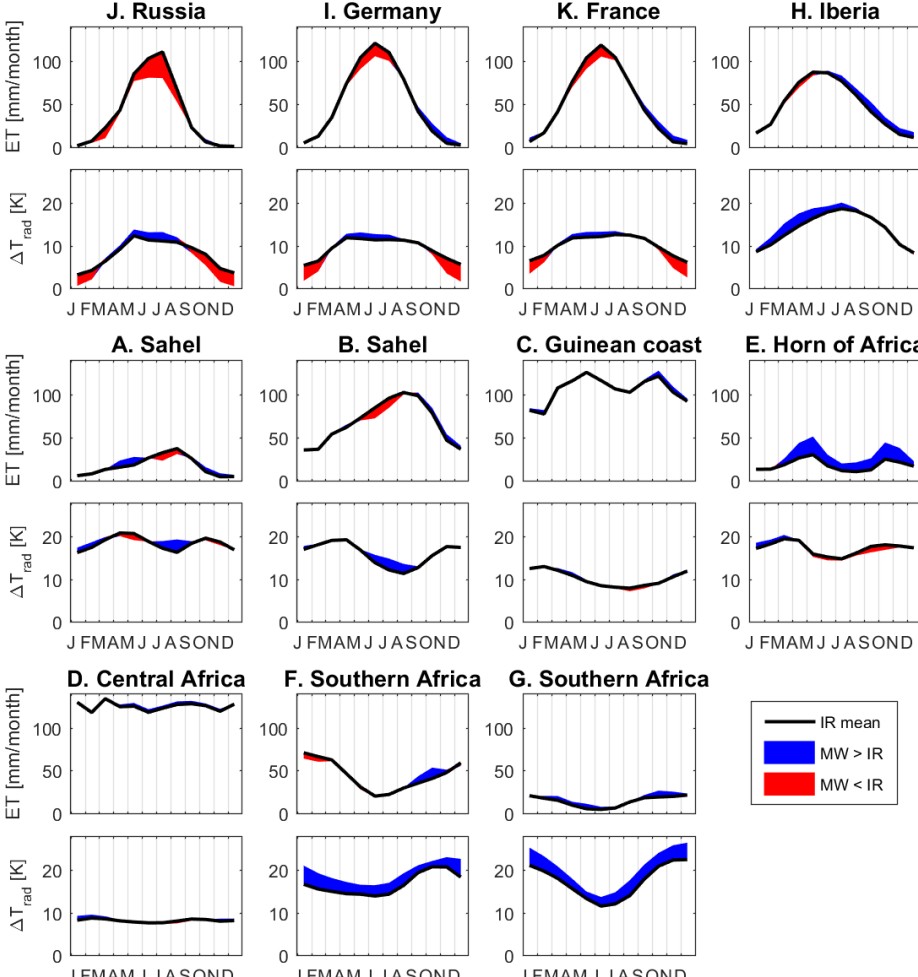

**Figure 5: Mean monthly ET as estimated with ALEXI-IR over 2003-2011, and monthly means of its MODIS-based $\Delta T_{rad}$ input (period 2004-2011), for selected regions. The deviation from these IR estimates when using the MW inputs is shown in blue for a positive deviation and red for a negative deviation.**











**Figure 6: As Fig 4, but now averaged by season.**

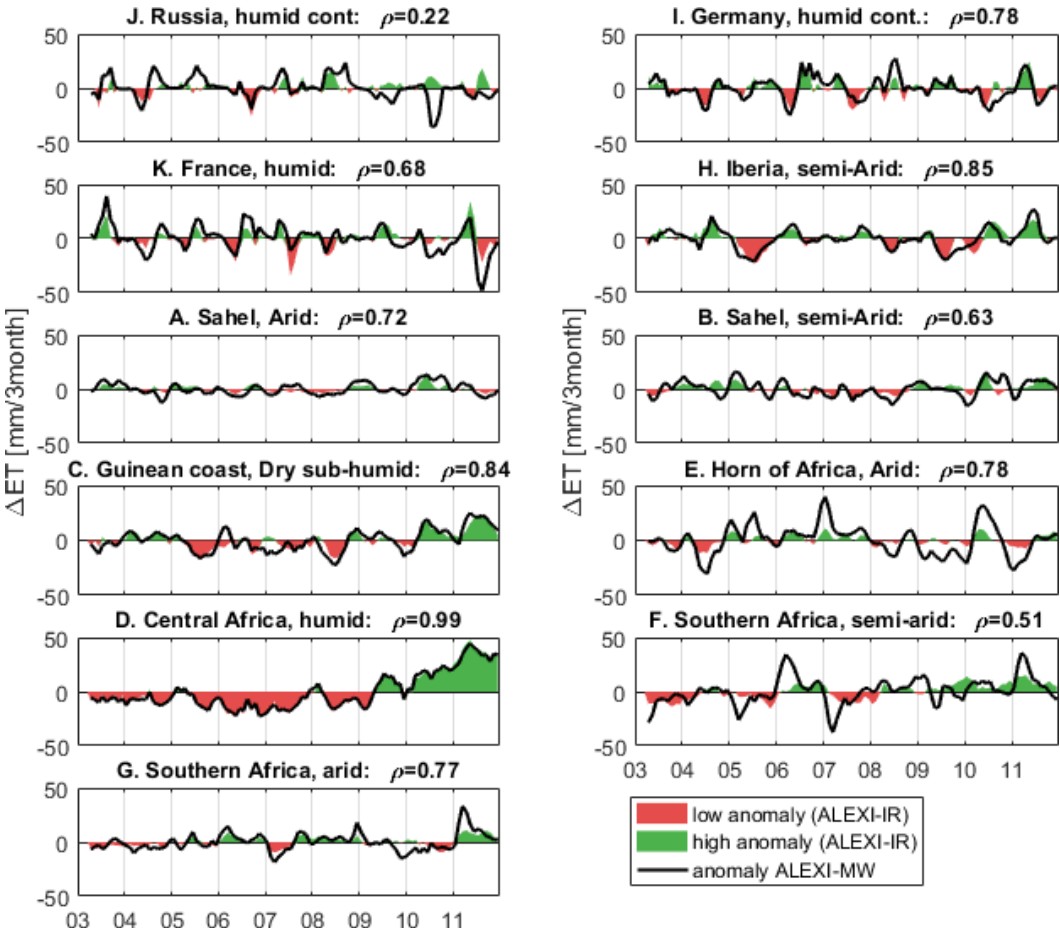

**Figure 7: Comparison of anomaly in 3-month ET totals as calculated from ALEXI-IR and ALEXI-MW for selected regions (see Fig. 3 for definition of regions).**





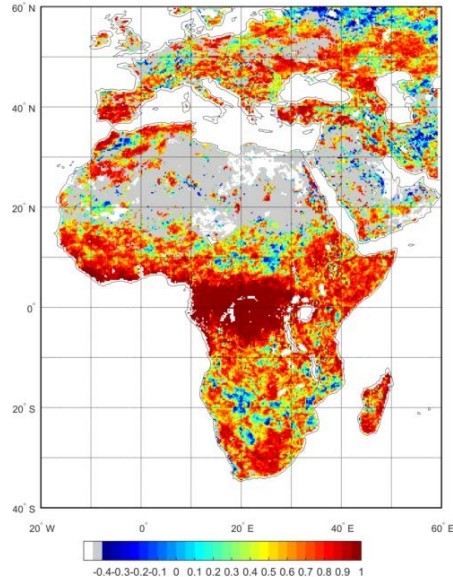

**Figure 8: Pearson's correlation between anomaly in 3-month ET totals as estimated by ALEXI-IR and ALEXI-MW. White areas have no data, grey areas are masked because the standard deviation in 3-month anomaly was below 17 mm/3month in both estimates.**





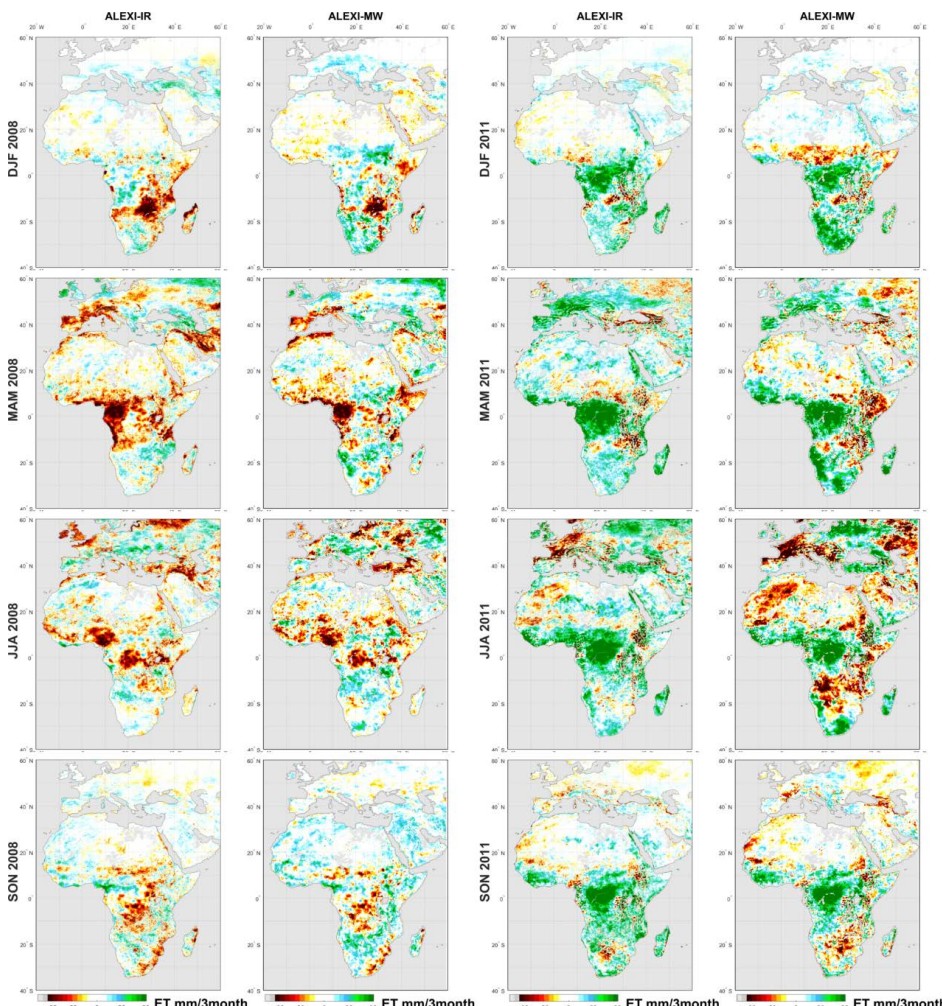

**Figure 9: Anomaly in seasonal ET compared to multi-year mean (2003-2011, see Fig 6) as retrieved by ALEXI-IR and ALEXI-MW. The first two columns show the anomalies for 2008 and the two right-hand columns show them for 2011.**



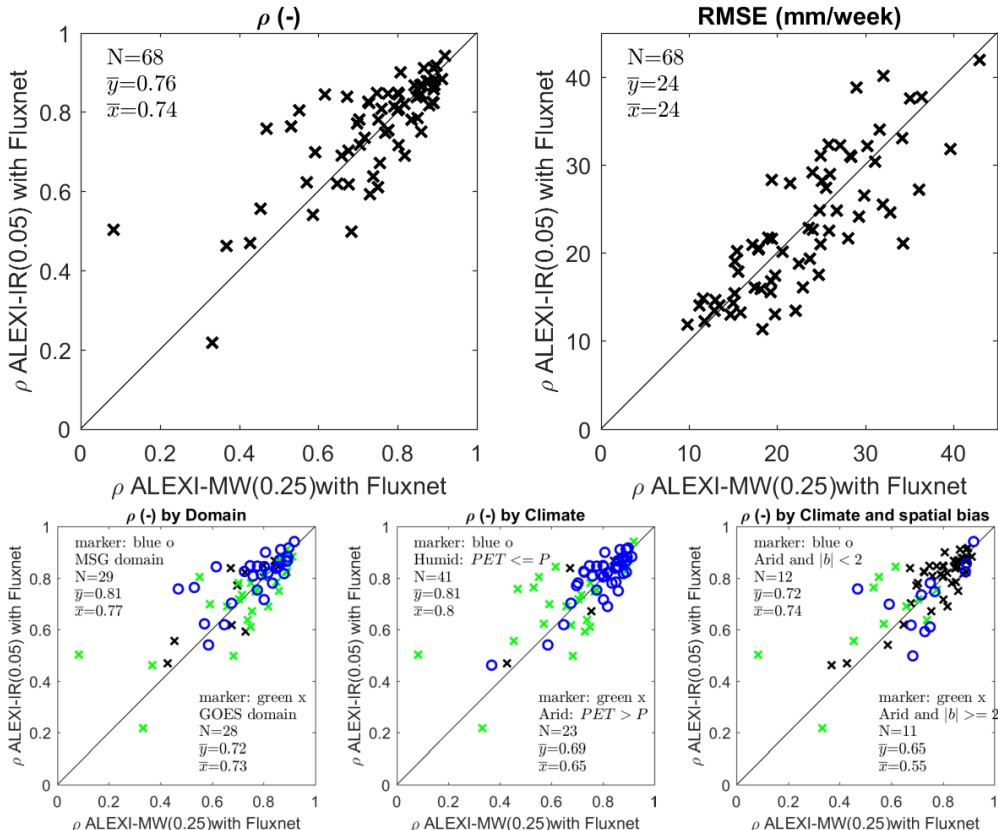

**Figure 10: Comparison of Pearson correlation (ρ) and RMSE between satellite data and Fluxnet observations. Top row: comparison between ALEXI-MW (X-axis) and ALEXI-IR (Y-axis). Second row: same data as presented in the left-hand panel on the top row, but now distinct subsets of the tower sites are emphasized. The first panel splits the sites by geographic region, the second panel based on climate (Humid Vs Arid, see text for definition). Panel three splits the 'arid' sites further based on bias between the ALEXI-IR (0.05°) and the mean value for the encompassing 0.25° grid box with a threshold of |b|=2mm/week. The black x mark stations that are either below 60°N, or are not covered by the two contrasting selections.**

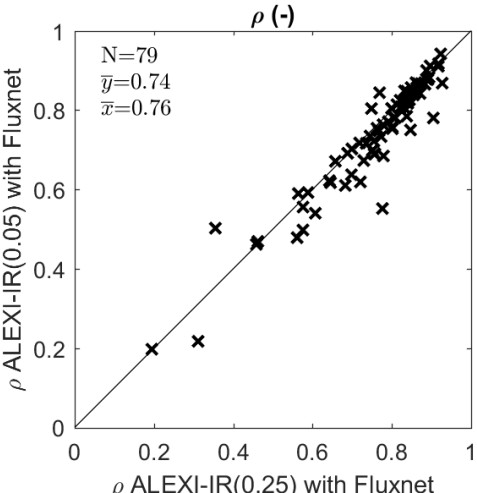

**Figure 11: Same as Fig 10, top left-hand panel, but now showing the effect of spatial resolution by comparing the Pearson correlation of the original ALEXI-IR(0.05°) with tower data (Y-axis), with that of ALEXI-IR(0.25°) and tower data (Y-axis).**





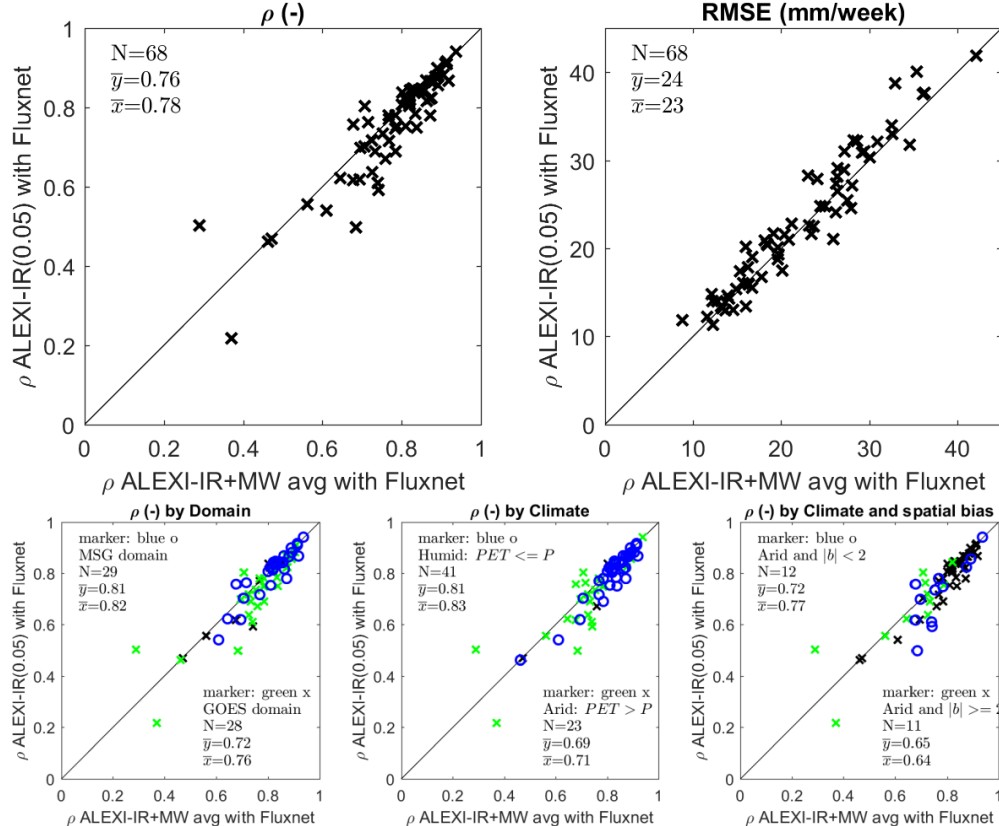

**Fig 12: Same as Fig 10, but now on the X-axis the temporal correspondence between Flux tower observations and ALEXI-IR+MW, the average of ALEXI-IR(0.05°) and ALEXI-MW(0.25°).**

