# Peer review of "Microwave implementation of two-source energy balance approach for estimating evapotranspiration"

_Hydrology and Earth System Sciences, 2017_

## Referee Comment (RC1) · C. Jimenez (Referee) · 11 Jul 2017

General comments

The paper presents an application of the ET modelling framework ALEXIS where the original TIR data is replaced by MW data. The performance of the new ALEXIS estimates are evaluated by comparison with the TIR based estimates, and by comparison with latent heat fluxes from a selection of tower sites. Overall, the paper is well written, the analysis of the presented results is adequate, and the figures illustrate well the major points of the analysis.

[Figure]

Intentionally, even if this is a MW based product, the authors have limited the analysis to the clear-sky estimates to facilitate the evaluation with the TIR estimates. I agree with this rationale, but I think the opportunity to already introduce some all-sky estimates in the tower evaluation was there. I suppose that even if there are some doubts about the boundary layer modelling for cloudy conditions, ALEXIS has been run for all available MW-LST Trads, so the ET outputs were available. If so, it would have been very interesting to see a tower evaluation of the 7-day totals of clear-sky ALEXI-TIR and ALEXI-MW, together with and ALEXI –MW 7-day totals but using all MW-Trad available. Presumably, estimating ET without the clear-sky restriction would have resulted in a better match to the tower observations, justifying all the trouble of developing a MW-based product. Given that using the MW Trad reduces the spatial resolution of the ET estimates by a factor of 5, there has to be some gain somewhere to justify their use.

I find adequate the level of description of ALEXIs in the text, but I'm missing some info about the concrete input datasets for this version of ALEXIS. Even if a paper presumably dealing with that is in review and cited in the text, at least the basic datasets to run ALEXIS in this global version should be given. The meteorology plays a crucial role in the derivation of the ALEXIS output, so I imagine that compared with the previous CONUS applications, something has been changed in that respect. A critical parameter there is the surface radiation, as the LST is also involved, and it may be of interest to know how the new LST data has been handle in that respect.

Specific comments

P1. L11. Given our current inability to properly validate global ET, I doubt that this exercise can be used to validate whether your diurnal temperature information from the MW obs is correct. You should show that your LST is reasonable before you attempt to estimate ET, as I think you did in previous publications, in which case there are no reasons for the ET to be unreasonable.

P1. L25. The limitation to clear-sky estimates applies to those ET methodologies relying in TIR data, but there are already all-sky global ET estimates not depending that clearly on TIR data, perhaps "all-sky LST-based estimates"?

P1. L28. The abstract should give information about the geographical coverage of the exercise, global?

P3. L11. Most ALEXI applications have been restricted in the past to the CONUS. Now, that it is starting to reach other domains, I wonder if there are some thoughts regarding an estimation of rain interception by the canopy, which can have some importance at some regions.

P3. L26. I think a few lines about the global implementation of ALEXI should be written. It is not clear if the global ALEXI is just plain ALEXI with global inputs, or something else has been changed. Also, what inputs ALEXI uses and the specific datasets are always interesting information. There are regions where Trad will have a very small impact on the ET estimates, i.e., there will be mainly driven by the meteorological data, so it is relevant to know about that data as well.

P3. L38. Is then the longwave heat loss recalculated with each temperature dataset? If so, how? I guess you need a daily integrated value for the Rnet, which is not straight-forward as the LST datasets are not geostationary this time.

P4. L6. And when not over the GOES disk, how is MODIS LST converted to a Delta-Trad?

P6. L1. It is not clear where the 7-day total comes from. I imagine there is a reason, linked to MODIS sampling (the MODIS ET product is 8-days). The meteorology can change quite significantly in 7 days, so if the 7-day total is based on one-day sampling, I can imagine that for some conditions the errors introduced can be large.

P6. L3. So the solar radiation is daily, and the ratio of ET to the radiation maintained constant for the 7-day period?

P6. L11. What is the reason for the missing 31% MW estimates when there are TIR estimates? The spatial sampling of the MW product?

P6. L12. Why should the MW-LST being calibrated to MODIS-LST?

P6. L25. Do you use the energy closure corrected fluxes or the original ones?

P6. L30. This is a bit confusing, even if discussed already in Section 2.3. If only in Africa and Europe the scaling of MW-LST supported, but you have then MW-LST globally, and presumably the ET, how "bad" is the MW-LST and ET outside those regions? Or, in other words, can we globally use the MW-LST ET or we need to wait for further developments in the MW-LST calibration outside those regions?

P7. L15 14 mm/month) THAN? Then?

P7. L18. I guess that's the grey color in Figure 4. It should be mentioned in the figure caption.

P8. L1. Have you found any explanations for the lack of agreement in the Horn of Africa? Trad seems to agree relatively well.

P8. L32. I'm surprised by a large part of France being masked by the grey color, i.e., not showing inter-annual variability. Most of the other grey areas are in the very dry regions, as expected, which is not the case of France.

P9. L14. Yes, very counterintuitive result and a bit worrying regarding the quality of these metrics. What do you mean by spatially uncorrelated noise? What MODIS? LST or the ET? This is not clear to me, it may be worth explaining better if you think you know the causes of this behavior.

P9. L20. What about the GOES-based time-extrapolation of the MODIS-LST to obtain Trad, can we see something here?

P9. L21. with AN alpha is?

P9. L24. Water limited regions are where Trad has a larger impact on the ET estimates. This suggests that ALEXI operates better when the ET depends less on Trad, right?

P10. L13. I think this needs a bit more of elaboration. Why do we want to merge both estimates? What is the overall strategy? The landscape is something that we cannot change, do you mean merging only when the landscape is homogeneous? But surely, merging spatial estimates at different resolutions will require up-scaling of the finest resolution one, or am I missing something?

P10. L15. I think is a bit too simplistic to just blame signal to noise ratio in arid regions as the reason for the poorer performance of ALEXIS there. Deriving the accurate stress needed for the water limited regions is where current ET methodologies, independent of modeling framework, tend to struggle, and ALEXI does not seem to be an exception here.

P11. L15. Why is this result surprising? If the MW-LST and TIR-LST are reasonable estimations of the true LST, and the rest of the ALEXIS modeling and inputs are unchanged, I will not expect large differences between both ALEXIs.

P11. L24. I would agree if ALEXI were using raw MW-LST and TIR-LST observations. But the MW-LST has been fitted to the TIR-LST, so I am not sure the independence is that obvious as stated in the text.

P11. L27. The improvement is 0.02, very modest, and it is obtained by combining estimates of different resolution, with one of them covering an area 5 times larger than the other. Given the discussion of the spatial degradation helping to improve correlations, I'm not sure we are just dealing with independence of datasets.

P11. L30. So far, all this is clear-sky. I suppose that MW-LST will be used always, not only for clear-sky, and that some work may be needed to see whether the assumptions of boundary layer development work for all-weather conditions. If so, it may be worth adding to this short roadmap.

---

## Referee Comment (RC2) · Anonymous Referee #2 · 11 Aug 2017

The authors have used diurnal temperature cycle built on available MW sensors to the well-known ALEXI model. The quality of diurnal temperature cycle based on MW sensors is important to ET retrieval. LSA-SAF LST was used to calibrate MW LST. I am wondering how did they determine or scale MW DTC parameters (especially the diurnal amplitude A) for the regions outside of SEVIRI coverage? Does MODIS ALEXI-IR also use LSA-SAF LST to determine DTC parameters?

I am confusing with eq. 1 and eq. 2 and 3. A_MW can be derived with equation 1. The diurnal cycle can then be produced with DTC3 model. eq. 2 says A_MW is scaled with TIR-based parameters. Then shouldn't the diurnal temperature cycle will also

be changed by the new A_MW? When A_MW equals A_IR, shouldn't dTrad_IR and dTrad_MW be the same or very close? Then ET_MW and ET_IR will certainly have a high correlation. Please compare with other ET dataset, such as the latitudinal transect in Fig. 6, Fig. 7, Fig. 9. Otherwise, comparison between ALEXI_IR and ALEXI_MW is not enough for evaluation of the method.

In addition, when they evaluate the ET results with Fluxnet, please do the analysis on a daily scale. A part of the purpose using MW here should be also providing ET at daily scale. So please assess the daily ET not weekly or monthly.

In the end, the authors have tried to fuse MW and IR ET. I am wondering why don't they use MW and IR signal to build diurnal temperature cycle directly. Could this method get more accurate daily global ET? Xuelong Chen has found MODIS monthly LST products could capture the monthly mean of diurnal LST variation. This means that the ALEXI could be used to MODIS monthly LST products. The authors might be interested to the following figure 1.

Eq1. If possible, please give the equation of DTC3, then the readers could quickly understand what kind of curves were used to fit the diurnal cycle.

Fig. 4 please specify t1 and t2 time for dTrad.

Fig. 8 how did the authors cope with different spatial resolution when they calculate Pearson's correlation. ALEXI-IR is 0.05 deg and ALEXI-MW is 0.25 deg. The correlation is at 0.05 deg resolution?

Fig. 9 why not calculate monthly anomaly? MW provide the possibility of daily ET. 3 months anomaly will provide a more consistent spatial patterns. But the performance at daily or monthly is more interesting. Fig. 7, and 8 also have the same question.

Figure. 10, when you calculate pearson correlation between satellite data and Fluxnet observations, daily, weekly, or monthly time series data was used? Please give RMSE at mm/day. This is more comparable to other's result.

[Figure]

page 2, 'generated a data record of weekly ET' why not daily? as above comments

Describe what is ti in Equation 1.

Table 2 Comparison is based on weekly averages in the period of 2003 to 2011. Why not use daily ET with gaps to calculate R, RMS? This is more useful for the readers to compare other ET products. Surely, weekly averages will give a higher R and low RMS. But MW provide ALEXI with the possibility for daily ET calculation.

Table 1, MOD43C3 doesn't have gaps? How did they fill albedo gaps? Please specify at what time step (00:00, 06:00. . .) lapse rate profile is used.
* * *
[Figure]

**Fig. 1.**

---

## Author Comment (AC1) · 15 Sep 2017

Please find below the comments of C. Jimenez and our replies.

Intentionally, even if this is a MW based product, the authors have limited the analysis to the clear-sky estimates to facilitate the evaluation with the TIR estimates. I agree with this rationale, but I think the opportunity to already introduce some all-sky estimates in the tower evaluation was there. I suppose that even if there are some doubts about the boundary layer modelling for cloudy conditions, ALEXI has been run for all available MW-LST Trads, so the ET outputs were available. If so, it would have been very interesting to see a tower evaluation of the 7-day totals of clear-sky ALEXI-TIR and

**HESSD**

ALEXI-MW, together with and ALEXI –MW 7-day totals but using all MW-Trad available. Presumably, estimating ET without the clear-sky restriction would have resulted in a better match to the tower observations, justifying all the trouble of developing a MW-based product. Given that using the MW Trad reduces the spatial resolution of the ET estimates by a factor of 5, there has to be some gain somewhere to justify their use.

I find adequate the level of description of ALEXIs in the text, but I'm missing some info about the concrete input datasets for this version of ALEXIS. Even if a paper presumably dealing with that is in review and cited in the text, at least the basic datasets to run ALEXIS in this global version should be given. The meteorology plays a crucial role in the derivation of the ALEXIS output, so I imagine that compared with the previous CONUS applications, something has been changed in that respect. A critical parameter there is the surface radiation, as the LST is also involved, and it may be of interest to know how the new LST data has been handle in that respect.

Reply. We thank the reviewer for the helpful comments. To the first question of why no preliminary investigating of all-sky estimates was included, the answer is that in this first-step paper we intentionally limited our scope to match the existing thermal-based ALEXI procedure. As a result, we did not process days that were flagged in the MODIS run, so we do not currently have an all-sky set available for this study. Investigation the behavior of ALEXI during more cloud covered conditions will be the topic of our future research. In terms of ET remote sensing, this paper is therefore a milestone along the way, not an end. In terms of MW-based LST remote sensing it is an important validation of independent information content compared to TIR products. The specific scope of the paper is now more clearly described in the abstract and introduction, as well as critical follow-on steps.

We also note the request for more information on ALEXI input datasets. While all input sets to ALEXI are listed in Table 1, we will add information on shortwave incoming radiation, which was omitted in the discussion paper. And more information on the calculation of LW fluxes is now included in response to detailed questions below.

P1. L11. Given our current inability to properly validate global ET, I doubt that this exercise can be used to validate whether your diurnal temperature information from the MW obs is correct. You should show that your LST is reasonable before you attempt to estimate ET, as I think you did in previous publications, in which case there are no reasons for the ET to be unreasonable.

Reply. In prior papers, we looked at temporal agreements between the satellite estimates and ground measurements from flux tower sites. What the analysis in this paper adds is a continental scale validation of the MW LST diurnal information using MODIS TIR LST. While the reviewer is correct that if the new LST inputs are reasonable, the ALEXI ET retrievals be reasonable, we move to the ET stage here to demonstrate sensitivity of this particular model to expected errors in the diurnal LST curve. The results show that the ET retrievals have some spatial differences that are explainable, but overall the TIR and MW ET estimates are sufficiently consistent under clear-sky conditions such that the next phases of investigation are warranted; namely evaluation of all-sky MW performance and development of a merged TIR-MW product.

P1. L25. The limitation to clear-sky estimates applies to those ET methodologies relying in TIR data, but there are already all-sky global ET estimates not depending that clearly on TIR data, perhaps "all-sky LST-based estimates"?

Reply. We narrowed our statement to say: ".. towards all-sky satellite-based retrieval of ET through an energy balance framework".

P1. L28. The abstract should give information about the geographical coverage of the exercise, global?

Reply. Indeed, 'global' will be added to the abstract.

P3. L11. Most ALEXI applications have been restricted in the past to the CONUS. Now, that it is starting to reach other domains, I wonder if there are some thoughts regarding an estimation of rain interception by the canopy, which can have some importance at

some regions.

Reply. The effect of rain water, evaporated from the canopy before it reached the soil, will have a cooling effect on the composite radiometric temperature. We hypothesize that the interception loss will be incorporated into the total ET retrieval via SEB, but may indeed be mistakenly attributed in the soil evaporation residual term. This is something we plan to test over LBA and related flux sites using the MW-LST ALEXI retrievals. Since accurate partitioning of ET may be required for some applications, pending results we plan work on approaches for proper attribution of the interception component. However, we prefer that ALEXI remains a purely diagnostic model to maintain its role as independent estimate of ET, not requiring precipitation data as input.

P3. L26. I think a few lines about the global implementation of ALEXI should be written. It is not clear if the global ALEXI is just plain ALEXI with global inputs, or something else has been changed. Also, what inputs ALEXI uses and the specific datasets are always interesting information. There are regions where Trad will have a very small impact on the ET estimates, i.e., there will be mainly driven by the meteorological data, so it is relevant to know about that data as well.

Reply. The description of the ALEXI model (Section 2.1) is improved to more clearly state the two relevant differences between the global implementation and previous geostationary implementations: 1) MODIS based Trad, 2) operation on 7-day averages.

P3. L38. Is then the longwave heat loss recalculated with each temperature dataset? If so, how? I guess you need a daily integrated value for the Rnet, which is not straightforward as the LST datasets are not geostationary this time.

Reply. Indeed, the LW heat loss is calculated in accordance with the temperature set that is used, using the Stephan-Boltzmann Law and a spatially varying emissivity. The ALEXI model computes the energy balance at two instantaneous points during the morning hours (post-dawn and pre-noon) using LST data available at those times. The latent heat estimate at the second time is then upscaled to a daily flux, conserving a

flux ratio metric. In prior applications, we used available energy as the scaling flux, and that did require an estimate of hourly LWup (clear and cloudy). In this case, the model estimates of canopy and soil temperatures at time 2 were tied to the diurnal air temperature curve as described in Sec. 2.4 of Anderson et al. (2012). However, especially for global applications we find we get similar yet more robust and computationally less expensive results by scaling using the insolation flux. This bypasses the need for hourly RNET/LST estimates. This distinction is made more clear by removing the reference to Rnet here, and replacing with turbulent fluxes: "There are two pathways through which Trad affects ALEXI ET estimates: directly through the estimation of the morning rise in temperature between time 1 and time 2, DTrad, which affects the boundary layer growth and the strength of the sensible heat fluxes; and indirectly through the calculation of the longwave heat loss, both at the canopy and at the soil surface, which affects the total energy available for the turbulent fluxes."

Reference added: Anderson, M.C., Kustas, W.P., Alfieri, J.G., Hain, C.R., Prueger, J.H., Evett, S.R., Colaizzi, P.D., Howell, T.A., & Chavez, J.L. (2012). Mapping daily evapotranspiration at Landsat spatial scales during the BEAREX'08 field campaign. Adv. Water Resour., 50, 162-177

P4. L6. And when not over the GOES disk, how is MODIS LST converted to a Delta-Trad?

Reply. We added a line to clarify this: "This regression model, trained over the GOES domain, is then applied globally to estimate Trad at time 1 and time 2 from MODIS LST."(P4. L11)

P6. L1. It is not clear where the 7-day total comes from. I imagine there is a reason, linked to MODIS sampling (the MODIS ET product is 8-days). The meteorology can change quite significantly in 7 days, so if the 7-day total is based on one-day sampling, I can imagine that for some conditions the errors introduced can be large.

Reply. The 7-day total comes from taking an average of all needed inputs on the "clearsky" days in the 7-day period and running ALEXI. Based on the resulting "clear sky" ET, we calculate a ratio of instantaneous latent heat flux to incoming solar radiation (fSUN). This fSUN is held constant over the 7-day period and used to calculate daily ET on each day, which is then summed to a 7-day total ET. This accounts for changes in atmospheric demand while preserving the evaporative fraction as determined on the clear-sky days. Running ALEXI on 7-day mean values was originally structured to limit computational demands and the system is currently being updated to produce daily simulations vs. the 7-day simulations used here.

P6. L3. So the solar radiation is daily, and the ratio of ET to the radiation maintained constant for the 7-day period?

Reply. This is correct. The daily solar radiation is used to account for changes in atmospheric demand while calculating the 7-day total ET.

P6. L11. What is the reason for the missing 31% MW estimates when there are TIR estimates? The spatial sampling of the MW product?

Reply. We added the following explanation in the text: "The reason this percentage is not higher is mainly due to the requirement of a near-noon overpass for the fitting of the diurnal temperature curve (See Section 2.3.2), in combination with the 1 in 3 days without such overpass for a given location as determined by the orbit and coverage of Aqua and GCOM-W."

P6. L12. Why should the MW-LST being calibrated to MODIS-LST?

Reply. We added the reason: ".. , something that is likely needed to maximize consistency between ALEXI implementations over the globe."( P6. L20.)

P6. L25. Do you use the energy closure corrected fluxes or the original ones?

Reply. This analysis made use of the gapfilled ones (LE_F_MDS), not the energy closure corrected fluxes, and is now specified in fluxnet data description section.

P6. L30. This is a bit confusing, even if discussed already in Section 2.3. If only in Africa and Europe the scaling of MW-LST supported, but you have then MW-LST globally, and presumably the ET, how "bad" is the MW-LST and ET outside those regions? Or, in other words, can we globally use the MW-LST ET or we need to wait for further developments in the MW-LST calibration outside those regions?

Reply. At the end of the paper we present the roadmap for global all-sky ET. The first step being: a global observation based calibration of MW-LST with MODIS-LST to reduce biases as identified at the high incidence angles of the MSG domain and avoid the need for extrapolation of scaling parameters.

P7. L15 14 mm/month) THAN? Then?

Reply. Corrected.

P7. L18. I guess that's the grey color in Figure 4. It should be mentioned in the figure caption. Reply. We added to the caption of Figure 4: '. . . .with areas with ET below 14 mm/month greyed out.'

P8. L1. Have you found any explanations for the lack of agreement in the Horn of Africa? Trad seems to agree relatively well.

Reply. The MW DTrad is a little bit lower than the TIR estimate, whereas the MW ET is much higher than the TIR estimate. The opposite signs are in agreement, but the response is larger than in other arid regions, like the Northern Sahel. We don't have an explanation for this higher sensitivity in this region at this point, but this is something we will be investigating. We hypothesize that the model be encountering a threshold condition in this region (e.g., running up against the constraint that throttles back the canopy transpiration when negative midday soil evaporation is obtained). This would make the model hypersensitive to small changes in DTrad in these areas.

P8. L32. I'm surprised by a large part of France being masked by the grey color, i.e., not showing inter-annual variability. Most of the other grey areas are in the very dry

regions, as expected, which is not the case of France.

Reply. Upon closer look at the creation of this figure an error was found in the processing. The figure in the original manuscript represent correlation in cumulative anomaly, not anomaly in 3-month totals. The new figures reflect the actual 1-month and 3-month cumulation of anomalies. And this indeed reveals France, as the reviewer expected it should.

P9. L14. Yes, very counterintuitive result and a bit worrying regarding the quality of these metrics. What do you mean by spatially uncorrelated noise? What MODIS? LST or the ET? This is not clear to me, it may be worth explaining better if you think you know the causes of this behavior.

Reply. The sentence itself can be clarified to an answer some of these questions: "This indicates the presence of spatially uncorrelated noise in the 0.05° MODIS LST input that is uncorrelated with the surrounding 0.25° grid average and negates any positive effect of its resolution advantage compared to a 0.25° grid average for most sites. ". It may be that the selected sites are in relatively homogeneous terrain in terms of ET, in which case statistics may be improved by averaging more pixels. It is clear though, that this is an issue that plays in the averaging from 0.05 to 0.25 ET retrievals. In the revised manuscript, we will split this out more clearly from the comparison between MW and TIR ET retrievals by using the 0.25 degree TIR ET as the reference. This raises the bar a little bit for MW ET in terms of correlation and has the same RMSE.

P9. L20. What about the GOES-based time-extrapolation of the MODIS-LST to obtain Trad, can we see something here?

Reply. This question refers to the lower left panel of Figure 10 where the Flux-tower results a broken down by domain, MSG vs GOES domain. On reflection, the difference we see is not easy to relate to domain for which the temperature sets were calibrated. While the paper L20 noted "This is despite the MODIS ALEXI-IR being calibrated with GOES data.", it seems more obvious that the overwhelming difference is between more

humid climates represented in the MSG domain as opposed to the GOES domain. As the number of datasets is too small to split them out by both domain and climate we likely will leave out this panel on 'domain' from the revised manuscript.

P9. L21. with AN alpha is?

Reply. Changed to 'an alpha parameter of 1.26'

P9. L24. Water limited regions are where Trad has a larger impact on the ET estimates. This suggests that ALEXI operates better when the ET depends less on Trad, right? Reply. Yes, in an extreme case, the humid tropics, the ALEXI ET is entirely driven by evaporative demand as parameterized with Priestly Taylor, with no impact of Trad. This is noted in Section 3.3. This is typically the case with most TIR-based ET models – when conditions are closer to potential, the model tends to perform better.

P10. L13. I think this needs a bit more of elaboration. Why do we want to merge both estimates? What is the overall strategy? The landscape is something that we cannot change, do you mean merging only when the landscape is homogeneous? But surely, merging spatial estimates at different resolutions will require up-scaling of the finest resolution one, or am I missing something?

Reply. I think the wording was not clear here. We changed to simply say: "The above analysis of flux tower data suggests that the prospect for merging MW and TIR is good for humid climate regions in general. In arid climates..." The overall strategy is outlined at the end of the conclusion.

P10. L15. I think is a bit too simplistic to just blame signal to noise ratio in arid regions as the reason for the poorer performance of ALEXIS there. Deriving the accurate stress needed for the water limited regions is where current ET methodologies, independent of modeling framework, tend to struggle, and ALEXI does not seem to be an exception here.

Reply. Signal to noise may indeed be only part of the explanation. We changed the

sentence to: "Partly, this reflects a lower signal to noise in areas with low seasonal variation in ET, but it also reflects a more challenging environment for ALEXI with little vegetation. "

P11. L15. Why is this result surprising? If the MW-LST and TIR-LST are reasonable estimations of the true LST, and the rest of the ALEXIS modeling and inputs are unchanged, I will not expect large differences between both ALEXIs.

Reply. We will remove the word 'surprisingly' and leave it to the reader to be surprised, or not. P11. L24. I would agree if ALEXI were using raw MW-LST and TIR-LST observations. But the MW-LST has been fitted to the TIR-LST, so I am not sure the independence is that obvious as stated in the text.

Reply. The mapping of MW brightness temperature to LST is based on 4 time-constant maps of fitting parameters – there is no transfer of temporal information from TIR-LST to MW-LST. P11. L27. The improvement is 0.02, very modest, and it is obtained by combining estimates of different resolution, with one of them covering an area 5 times larger than the other. Given the discussion of the spatial degradation helping to improve correlations, I'm not sure we are just dealing with independence of datasets.

Reply. We agree that there may be more factors at play with this limited set of flux towers. The sentence is changed to a less strong wording: "The overall observed improvement from rho=0.76 to rho=0.78 confirms is in agreement with an expectation that part of the error in ET estimates from ALEXI-IR and ALEXI-MW is independent. "

P11. L30. So far, all this is clear-sky. I suppose that MW-LST will be used always, not only for clear-sky, and that some work may be needed to see whether the assumptions of boundary layer development work for all-weather conditions. If so, it may be worth adding to this short roadmap.

Reply. Indeed, this was an omission in our roadmap. We added a line in the conclusion: "For clear skies, it appears that a simple averaging of ALEXI-IR and ALEXI-MW

would provide for a reduction in estimation uncertainty of ALEXI ET for times when both are available. Finally, the all-sky implementation that is now within reach with ALEXI-MW will require careful investigation of the assumptions related to the boundary layer development under less stable conditions."

---

## Author Comment (AC2) · 15 Sep 2017

Please find below the comments of Anonymous Referee #2 and our replies.

The authors have used diurnal temperature cycle built on available MW sensors to the well-known ALEXI model. The quality of diurnal temperature cycle based on MW sensors is important to ET retrieval. LSA-SAF LST was used to calibrate MW LST.

I am wondering how did they determine or scale MW DTC parameters (especially the diurnal amplitude A) for the regions outside of SEVIRI coverage? Does MODIS ALEXI IR also use LSA-SAF LST to determine DTC parameters? I am confusing with eq.

1 and eq. 2 and 3. A_MW can be derived with equation 1. The diurnal cycle can then be produced with DTC3 model. eq. 2 says A_MW is scaled with TIR-based parameters. Then shouldn't the diurnal temperature cycle will also be changed by the new A_MW? When A_MW equals A_IR, shouldn't dTrad_IR and dTrad_MW be the same or very close? Then ET_MW and ET_IR will certainly have a high correlation. Please compare with other ET dataset, such as the latitudinal transect in Fig. 6, Fig. 7, Fig. 9. Otherwise, comparison between ALEXI_IR and ALEXI_MW is not enough for evaluation of the method. In addition, when they evaluate the ET results with Fluxnet, please do the analysis on a daily scale. A part of the purpose using MW here should be also providing ET at daily scale. So please assess the daily ET not weekly or monthly.

Reply. Thank you for the comments and reflection on the manuscript. In response to these general comments we want to stress that:

- Only the long term mean of the diurnal parameters is scaled to match those of LSA SAF TIR. The entire transfer of information from TIR to MW is contained in three static maps (for amplitude, mean, and timing). This means that the correlation between the diurnal amplitudes of TIR and MW LST is not affected – in terms of short-scale inter-seasonal patterns the two sets remain independent. This point is will be emphasized in the description of Eq2-3 and the process of creating MW-LST: Section 2.3.3: "Because all three parameters are constant with time, Eqs 2-3 preserve their temporal independence of the TIR LST product."

- For the extrapolation of MW to LSA-SAF LST calibration parameters outside the Meteosat domain we used ad-hoc linear regressions with vegetation characteristics (MW vegetation optical depth). It is not an approach we will continue with. For this paper, we use it to bring the dataset in the approximate range of the TIR LST, but only to include it in the analysis of temporal signal with Fluxnet data. Again, the temporal signal is not affected by the calibration.

In light of this, we do think a comparison between MW and TIR ALEXI is a valid test of

performance at this point and a strong indication of diurnal information content in MW LST.

I am wondering why don't they use MW and IR signal to build diurnal temperature cycle directly. Could this method get more accurate daily global ET? Xuelong Chen has found MODIS monthly LST products could capture the monthly mean of diurnal LST variation. This means that the ALEXI could be used to MODIS monthly LST products. The authors might be interested to the following figure 1.

Reply. It is certainly possible that we will eventually merge MW and IR to get a better estimate of diurnal temperature. However, this still requires some additional steps to overcome, not least of which is the spatial resolution difference. This paper presents a milestone along the way, reporting on an experiment to test MW LST in an ET retrieval that is itself sensitive to the diurnal information.

Eq1. If possible, please give the equation of DTC3, then the readers could quickly understand what kind of curves were used to fit the diurnal cycle.

Reply. We will add the description of the DTC model as a model that " combines a cosine and an exponential term to describe the effect of the sun and the decrease of surface temperature at night". We think that this level of detail is sufficient for understanding the present paper, and listing the exact equations might distract from the main analysis. For a detailed description, the reader is referred to Holmes et al 2015 and references therein.

Section 2.3.2 now includes this passage: "The DTC model combines a cosine and an exponential term to describe the effect of the sun and the decrease of surface temperature at night and is based on Göttsche and Olesen (2001) with slight adaptations to limit the number of parameters. This implementation (DTC3) is fully described in Holmes et al. (2015). "

Fig. 4 please specify t1 and t2 time for dTrad.

Reply. We added to the caption: dTrad is defined in the ALEXI framework as the temperature rise between 1.5 hr after sunrise to 1.5 hr before noon (see Section 2.1).

Fig. 8 how did the authors cope with different spatial resolution when they calculate Pearson's correlation. ALEXI-IR is 0.05 deg and ALEXI-MW is 0.25 deg. The correlation is at 0.05 deg resolution?

Reply. The correlation in 3-month anomalies (Fig 8) is calculated based on 0.25 degree resolution data. This is now specified in the caption.

Fig. 9 why not calculate monthly anomaly? MW provide the possibility of daily ET. 3 months anomaly will provide more consistent spatial patterns. But the performance at daily or monthly is more interesting. Fig. 7, and 8 also have the same question.

Reply. 3-month anomalies were chosen to represent the larger seasonal patterns in interannual variation. Higher temporal resolution is tested with in situ data's. As can be seen in the attached Figure 1, the correlations are rather similar between the 1-month and 3-month anomalies.

Reply. Thanks for pointing this out, the caption did not specify that these statistics are based on weekly ET values. For ALEXI these weekly values are interpolated from clearsky observations as detailed in Section 2.4. The Fig 10 caption is changed to start with: "Comparison of Pearson correlation () and RMSE between estimates of weekly ET from satellite data and Fluxnet observations."

Describe what is ti in Equation 1.

Reply. The definition of t is given in the preceding sentence, 'ti' is the observation time of the temperature measurement. The index 'i' is used to represent the individual observations within a day.

Table 2 Comparison is based on weekly averages in the period of 2003 to 2011. Why not use daily ET with gaps to calculate R, RMS? This is more useful for the readers to compare other ET products. Surely, weekly averages will give a higher R and low

RMS. But MW provide ALEXI with the possibility for daily ET calculation. AND page 2, 'generated a data record of weekly ET' why not daily? as above comments \

Reply. While the goal of this work is to eventually reduce the temporal interval of consistent ET retrievals to daily, at this moment we followed the existing global ALEXI protocol that allows for evaluation of weekly ET totals. We do look forward to analyzing daily output in future studies, but it was not an option at this stage of the development.

Table 1, MOD43C3 doesn't have gaps? How did they fill albedo gaps? Please specify at what time step (00:00, 06:00: : :) lapse rate profile is used.

Reply. In fact, it should have read MCD43B3 and this is a 16-day product. The updated reference (Schaaf et al. 2002) explains: ".. BRDF parameters are produced (via either full or magnitude inversions) for every land or coastal area which is viewed (and atmospherically corrected) at least once over a 16-day period. Land areas that remain completely cloud covered over this period are designated with fill values."

Schaaf, C. B., Gao, F., Strahler, A. H., Lucht, W., Li, X., Tsang, T., Strugnell, N. C., Zhang, X., Jin, Y., Muller, J.-P., Lewis, P., Barnsley, M., Hobson, P., Disney, M., Roberts, G., Dunderdale, M., Doll, C., d'Entremont, R. P., Hu, B., Liang, S., Privette, J. L. and Roy, D.: First operational BRDF, albedo nadir reflectance products from MODIS, Remote Sens. Environ., 83(1–2), 135–148, doi:10.1016/S0034-4257(02)00091-3, 2002.

[Figure]

Figure with two maps of Africa showing Pearson's correlation.

**Fig. 1.** Figure 8: Pearson's correlation between anomaly in 1-month (left) and 3-month (right) ET totals as estimated by ALEXI-IR and ALEXI-MW, calculated at 0.25 degree resolution. White areas have no data, g

---

## Author Response (AR1)

Dear Editor,

Please find below the point-by-point reply to the reviewer's comments (in italic) and our replies (preceded by "»"). The most relevant changes to the manuscript are a reconsidered presentation of the comparison with flux tower observations. We now isolate the spatial resolution effect in a first step and then compare MW and TIR ALEXI both at the lower resolution of MW data. We also removed the test of a simple combination of the two methods, as this does not improve it relative to this baseline and will need more thorough analysis. We also detailed the roadmap towards a combined MW and TIR ALEXI more clearly.

We look forward to hearing your opinion on this revised manuscript.

On behalf of all authors,

Thomas Holmes

**Reviewer 1.**

*Intentionally, even if this is a MW based product, the authors have limited the analysis to the clear-sky estimates to facilitate the evaluation with the TIR estimates. I agree with this rationale, but I think the opportunity to already introduce some all-sky estimates in the tower evaluation was there. I suppose that even if there are some doubts about the boundary layer modelling for cloudy conditions, ALEXIS has been run for all available MW-LST Trads, so the ET outputs were available. If so, it would have been very interesting to see a tower evaluation of the 7-day totals of clear-sky ALEXI-TIR and ALEXI-MW, together with and ALEXI –MW 7-day totals but using all MW-Trad available. Presumably, estimating ET without the clear-sky restriction would have resulted in a better match to the tower observations, justifying all the trouble of developing a MW-based product. Given that using the MW Trad reduces the spatial resolution of the ET estimates by a factor of 5, there has to be some gain somewhere to justify their use.*

*I find adequate the level of description of ALEXIs in the text, but I'm missing some info about the concrete input datasets for this version of ALEXIS. Even if a paper presumably dealing with that is in review and cited in the text, at least the basic datasets to run ALEXIS in this global version should be given. The meteorology plays a crucial role in the derivation of the ALEXIS output, so I imagine that compared with the previous CONUS applications, something has been changed in that respect. A critical parameter there is the surface radiation, as the LST is also involved, and it may be of interest to know how the new LST data has been handle in that respect.*

» We thank the reviewer for the helpful comments.

To the first question of why no preliminary investigating of all-sky estimates was included, the answer is that in this first-step paper we intentionally limited our scope to match the existing thermal-based ALEXI procedure. As a result, we did not process days that were flagged in the MODIS run, so we do not currently have an all-sky set available for this study. Investigation the behavior of ALEXI during more cloud covered conditions will be the topic of our future research. In terms of ET remote sensing, this paper is therefore a milestone along the way, not an end. In terms of MW-based LST remote sensing it is an important validation of independent information content compared to TIR products.  The specific

scope of the paper is now more clearly described in the abstract and introduction, as well as critical follow-on steps.

We also note the request for more information on ALEXI input datasets. While all input sets to ALEXI are listed in Table 1, we will add information on shortwave incoming radiation, which was omitted in the discussion paper. And more information on the calculation of LW fluxes is now included in response to detailed questions below.

*P1. L11. Given our current inability to properly validate global ET, I doubt that this exercise can be used to validate whether your diurnal temperature information from the MW obs is correct. You should show that your LST is reasonable before you attempt to estimate ET, as I think you did in previous publications, in which case there are no reasons for the ET to be unreasonable.*
» In prior papers, we looked at temporal agreements between the satellite estimates and ground measurements from flux tower sites.  What the analysis in this paper adds is a continental scale validation of the MW LST diurnal information using MODIS TIR LST. While the reviewer is correct that if the new LST inputs are reasonable, the ALEXI ET retrievals be reasonable, we move to the ET stage here to demonstrate sensitivity of this particular model to expected errors in the diurnal LST curve.  The results show that the ET retrievals have some spatial differences that are explainable, but overall the TIR and MW ET estimates are sufficiently consistent under clear-sky conditions such that the next phases of investigation are warranted; namely evaluation of all-sky MW performance and development of a merged TIR-MW product.

*P1. L25. The limitation to clear-sky estimates applies to those ET methodologies relying in TIR data, but there are already all-sky global ET estimates not depending that clearly on TIR data, perhaps "all-sky LST-based estimates"?*
» We narrowed our statement to say: ".. towards all-sky satellite-based retrieval of ET through an energy balance framework".

*P1. L28. The abstract should give information about the geographical coverage of the exercise, global?*
» Indeed, 'global' is now added to the abstract (L17).

*P3. L11. Most ALEXI applications have been restricted in the past to the CONUS. Now, that it is starting to reach other domains, I wonder if there are some thoughts regarding an estimation of rain interception by the canopy, which can have some importance at some regions.*
» The effect of rain water, evaporated from the canopy before it reached the soil, will have a cooling effect on the composite radiometric temperature. We hypothesize that the interception loss will be incorporated into the total ET retrieval via SEB, but may indeed be mistakenly attributed in the soil evaporation residual term. This is something we plan to test over LBA and related flux sites using the MW-LST ALEXI retrievals. Since accurate partitioning of ET may be required for some applications, pending results we plan work on approaches for proper attribution of the interception component. However, we prefer that ALEXI remains a purely diagnostic model to maintain its role as independent estimate of ET, not requiring precipitation data as input. In the revised manuscript we make a note of this in the roadmap outlined in the discussion: "Finally, the all-sky implementation that is now within reach with ALEXI-MW will test the assumptions in new ways, which will require careful investigation. For example, the assumptions related to the boundary layer development may be tested as we move to include less stable conditions associated with cloudy skies. Similarly, evaporation of intercepted rain

water will feature more prominently under cloudy skies and may require inclusion as a separate process within the current physical framework.".

*P3. L26. I think a few lines about the global implementation of ALEXI should be written. It is not clear if the global ALEXI is just plain ALEXI with global inputs, or something else has been changed. Also, what inputs ALEXI uses and the specific datasets are always interesting information. There are regions where Trad will have a very small impact on the ET estimates, i.e., there will be mainly driven by the meteorological data, so it is relevant to know about that data as well.*
» The description of the ALEXI model (Section 2.1) is improved to more clearly state the two relevant differences between the global implementation and previous geostationary implementations: 1) MODIS based Trad, 2) operation on 7-day averages.

*P3. L38. Is then the longwave heat loss recalculated with each temperature dataset? If so, how? I guess you need a daily integrated value for the Rnet, which is not straightforward as the LST datasets are not geostationary this time.*
» Indeed, the LW heat loss is calculated in accordance with the temperature set that is used, using the Stephan-Boltzmann Law and a spatially varying emissivity. The ALEXI model computes the energy balance at two instantaneous points during the morning hours (post-dawn and pre-noon) using LST data available at those times. The latent heat estimate at the second time is then upscaled to a daily flux, conserving a flux ratio metric.  In prior applications, we used available energy as the scaling flux, and that did require an estimate of hourly LWup (clear and cloudy).  In this case, the model estimates of canopy and soil temperatures at time 2 were tied to the diurnal air temperature curve as described in Sec. 2.4 of Anderson et al. (2012).   However, especially for global applications we find we get similar yet more robust and computationally less expensive results by scaling using the insolation flux.  This bypasses the need for hourly Rnet/LST estimates.

This distinction is made more clear by stating that the Rnet is only calculated at time 1 and 2: "There are two pathways through which the T_rad input affects ALEXI ET estimates: through the estimation of the morning rise in temperature between time 1 and time 2, ΔT_rad, which affects the boundary layer growth and the strength of the sensible heat fluxes; and through the impact of TRAD on the upwelling longwave component of R_net at these times. "

Reference added: Anderson, M.C., Kustas, W.P., Alfieri, J.G., Hain, C.R., Prueger, J.H., Evett, S.R., Colaizzi, P.D., Howell, T.A., & Chavez, J.L. (2012). Mapping daily evapotranspiration at Landsat spatial scales during the BEAREX'08 field campaign. Adv. Water Resour., 50, 162-177

*P4. L6. And when not over the GOES disk, how is MODIS LST converted to a Delta- Trad?*
» We added a line to clarify this: "This regression model, trained over the GOES domain, is then applied globally to estimate Trad at time 1 and time 2 from MODIS LST."(Section 2.2)

*P6. L1. It is not clear where the 7-day total comes from. I imagine there is a reason, linked to MODIS sampling (the MODIS ET product is 8-days). The meteorology can change quite significantly in 7 days, so if the 7-day total is based on one-day sampling, I can imagine that for some conditions the errors introduced can be large.*
» The 7-day total comes from taking an average of all needed inputs on the "clear-sky" days in the 7-day period and running ALEXI. Based on the resulting "clear sky" ET, we calculate a ratio of instantaneous latent heat flux to incoming solar radiation (fSUN). This fSUN is held constant over the 7-day period and

used to calculate daily ET on each day, which is then summed to a 7-day total ET. This accounts for changes in atmospheric demand while preserving the evaporative fraction as determined on the clear-sky days.  Running ALEXI on 7-day mean values was originally structured to limit computational demands and the system is currently being updated to produce daily simulations vs. the 7-day simulations used here. This information is added to Section 2.1 of the revised manuscript.

*P6. L3. So the solar radiation is daily, and the ratio of ET to the radiation maintained constant for the 7-day period?*
» This is correct. The daily solar radiation is used to account for changes in atmospheric demand while calculating the 7-day total ET (see response above).

*P6. L11. What is the reason for the missing 31% MW estimates when there are TIR estimates? The spatial sampling of the MW product?*
» We added the following explanation in the text in section 2.3.5: "The reason this percentage is not higher is mainly due to the requirement of a near-noon overpass for the fitting of the diurnal temperature curve (See Section 2.3.2), in combination with the 1 in 3 days without such overpass for a given location as determined by the orbit and coverage of Aqua and GCOM-W."

*P6. L12. Why should the MW-LST being calibrated to MODIS-LST?*
» We added the reason: ".. , something that is likely needed to maximize consistency between ALEXI implementations over the globe."( P6. L28.)

*P6. L25. Do you use the energy closure corrected fluxes or the original ones?*
» This analysis made use of the gapfilled ones (LE_F_MDS), not the energy closure corrected fluxes, and is now specified in section 2.4.

*P6. L30. This is a bit confusing, even if discussed already in Section 2.3. If only in Africa and Europe the scaling of MW-LST supported, but you have then MW-LST globally, and presumably the ET, how "bad" is the MW-LST and ET outside those regions? Or, in other words, can we globally use the MW-LST ET or we need to wait for further developments in the MW-LST calibration outside those regions?*
» At the end of the revised paper (p12 L9) we present the roadmap for global all-sky ET. The first step being: a global observation based calibration of MW-LST with MODIS-LST to reduce biases as identified at the high incidence angles of the MSG domain and avoid the need for extrapolation of scaling parameters.

*P7. L15 14 mm/month) THAN? Then?*
» Corrected.

*P7. L18. I guess that's the grey color in Figure 4. It should be mentioned in the figure caption.*
» We added to the caption of Figure 4: '. …with areas with ET below 14 mm/month greyed out.'

*P8. L1. Have you found any explanations for the lack of agreement in the Horn of Africa? Trad seems to agree relatively well.*
» The MW ΔTrad is a little bit lower than the TIR estimate, whereas the MW ET is much higher than the TIR estimate. The opposite signs are in agreement, but the response is larger than in other arid regions, like the Northern Sahel. We don't have an explanation for this higher sensitivity in this region at this point, but this is something we will be investigating.  We hypothesize that the model be encountering a threshold condition in this region (e.g., running up against the constraint that throttles back the canopy

transpiration when negative midday soil evaporation is obtained). This would make the model hypersensitive to small changes in DT in these areas.

*P8. L32. I'm surprised by a large part of France being masked by the grey color, i.e., not showing inter-annual variability. Most of the other grey areas are in the very dry regions, as expected, which is not the case of France.*
» Upon closer look at the creation of this figure an error was found in the processing. The figure in the original manuscript represent correlation in cumulative anomaly, not anomaly in 3-month totals. The new figures reflect the actual 1-month and 3-month cumulation of anomalies. And this indeed reveals France, as the reviewer expected it should.

*P9. L14. Yes, very counterintuitive result and a bit worrying regarding the quality of these metrics. What do you mean by spatially uncorrelated noise? What MODIS? LST or the ET? This is not clear to me, it may be worth explaining better if you think you know the causes of this behavior.*
» The sentence itself (*P9. L33*) can be clarified to answer some of these questions: "This indicates the presence of spatially uncorrelated noise in the 0.05° MODIS LST input that is uncorrelated with the surrounding 0.25° grid average and negates any positive effect of its resolution advantage compared to a 0.25° grid average for most sites. ". It may be that the selected sites are in relatively homogeneous terrain in terms of ET, in which case statistics may be improved by averaging more pixels. It is clear though, that this is an issue that plays in the averaging from 0.05 to 0.25 ET retrievals. In the revised manuscript, we will split this out more clearly from the comparison between MW and TIR ET retrievals by using the 0.25 degree TIR-ET as the reference. This raises the bar a little bit for MW ET in terms of correlation but has the same RMSE.

*P9. L20. What about the GOES-based time-extrapolation of the MODIS-LST to obtain Trad, can we see something here?*
» It seems more obvious that the overwhelming difference is between more humid climates represented in the MSG domain as opposed to the GOES domain. As the number of datasets is too small to split them out by both domain and climate we cannot investigate the effect of calibration domain further.

*P9. L21. with AN alpha is?*
» Changed to 'an alpha parameter of 1.26'

*P9. L24. Water limited regions are where Trad has a larger impact on the ET estimates. This suggests that ALEXI operates better when the ET depends less on Trad, right?*
» Yes, in an extreme case, the humid tropics, the ALEXI ET is entirely driven by evaporative demand as parameterized with Priestly Taylor, with no impact of Trad. This is noted in Section 3.3. This is typically the case with most TIR-based ET models – when conditions are closer to potential, the model tends to perform better.

*P10. L13. I think this needs a bit more of elaboration. Why do we want to merge both estimates? What is the overall strategy? The landscape is something that we cannot change, do you mean merging only when the landscape is homogeneous? But surely, merging spatial estimates at different resolutions will require up-scaling of the finest resolution one, or am I missing something?*
» I think the wording was not clear here. We changed to simply say: "The above analysis of flux tower

data suggests that the prospect for merging MW and TIR is good for humid climate regions in general. In arid climates…" The overall strategy is outlined at the end of the conclusion.

*P10. L15. I think is a bit too simplistic to just blame signal to noise ratio in arid regions as the reason for the poorer performance of ALEXIS there. Deriving the accurate stress needed for the water limited regions is where current ET methodologies, independent of modeling framework, tend to struggle, and ALEXI does not seem to be an exception here.*
» Signal to noise may indeed be only part of the explanation. We changed the sentence (start of 3.5) to: "Partly, this reflects a lower signal to noise in areas with low seasonal variation in ET, but it also reflects a more challenging environment for ALEXI with little vegetation. "

*P11. L15. Why is this result surprising? If the MW-LST and TIR-LST are reasonable estimations of the true LST, and the rest of the ALEXIS modeling and inputs are unchanged, I will not expect large differences between both ALEXIs.*
» We removed the word 'surprisingly' and leave it to the reader to be surprised, or not.

*P11. L24. I would agree if ALEXI were using raw MW-LST and TIR-LST observations. But the MW-LST has been fitted to the TIR-LST, so I am not sure the independence is that obvious as stated in the text.*
» The mapping of MW brightness temperature to LST is based on 4 time-constant maps of fitting parameters – there is no transfer of temporal information from TIR-LST to MW-LST.

*P11. L27. The improvement is 0.02, very modest, and it is obtained by combining estimates of different resolution, with one of them covering an area 5 times larger than the other. Given the discussion of the spatial degradation helping to improve correlations, I'm not sure we are just dealing with independence of datasets.*
» We agree that there may be more factors at play with this limited set of flux towers. The sentence (P12. L6) is changed to a less strong wording: "The overall observed improvement from ρ=0.76 to ρ=0.78 confirms is in agreement with an expectation that part of the error in ET estimates from ALEXI-IR and ALEXI-MW is independent. "

*P11. L30. So far, all this is clear-sky. I suppose that MW-LST will be used always, not only for clear-sky, and that some work may be needed to see whether the assumptions of boundary layer development work for all-weather conditions. If so, it may be worth adding to this short roadmap.*
» Indeed, this was an omission in our roadmap. We added a few lines in the conclusion: "Finally, the all-sky implementation that is now within reach with ALEXI-MW will test the assumptions in new ways, which will require careful investigation. For example, the assumptions related to the boundary layer development may be tested as we move to include less stable conditions associated with cloudy skies. Similarly, evaporation of intercepted rain water will feature more prominently under cloudy skies and may require inclusion as a separate process within the current physical framework.".

**Reviewer 2.**

*The authors have used diurnal temperature cycle built on available MW sensors to the well-known ALEXI model. The quality of diurnal temperature cycle based on MW sensors is important to ET retrieval. LSA-SAF LST was used to calibrate MW LST.*

*I am wondering how did they determine or scale MW DTC parameters (especially the diurnal amplitude A) for the regions outside of SEVIRI coverage? Does MODIS ALEXI IR also use LSA-SAF LST to determine DTC parameters? I am confusing with eq. 1 and eq. 2 and 3. A_MW can be derived with equation 1. The diurnal cycle can then be produced with DTC3 model. eq. 2 says A_MW is scaled with TIR-based parameters. Then shouldn't the diurnal temperature cycle will also be changed by the new A_MW? When A_MW equals A_IR, shouldn't dTrad_IR and dTrad_MW be the same or very close? Then ET_MW and ET_IR will certainly have a high correlation. Please compare with other ET dataset, such as the latitudinal transect in Fig. 6, Fig. 7, Fig. 9. Otherwise, comparison between ALEXI_IR and ALEXI_MW is not enough for evaluation of the method.*

*In addition, when they evaluate the ET results with Fluxnet, please do the analysis on a daily scale. A part of the purpose using MW here should be also providing ET at daily scale. So please assess the daily ET not weekly or monthly.*

» Thank you for the comments and reflection on the manuscript. In response to these general comments we want to stress that:

-        Only the long term mean of the diurnal parameters is scaled to match those of LSA SAF TIR. The entire transfer of information from TIR to MW is contained in three static maps (for amplitude, mean, and timing). This means that the correlation between the diurnal amplitudes of TIR and MW LST is not affected – in terms of short-scale inter-seasonal patterns the two sets remain independent.

This point is now emphasized in the description of Eq2-3 and the process of creating MW-LST:

Section 2.3.3: "Because all three parameters are constant with time, Eqs 2-3 preserve their temporal independence of the TIR LST product."

-        For the extrapolation of MW to LSA-SAF LST calibration parameters outside the Meteosat domain we used ad-hoc linear regressions with vegetation characteristics (MW vegetation optical depth). It is not an approach we will continue with. For this paper, we use it to bring the dataset in the approximate range of the TIR LST, but only to include it in the analysis of temporal signal with Fluxnet data. Again, the temporal signal is not affected by the calibration.

In light of this, we do think a comparison between MW and TIR ALEXI is a valid test of performance at this point and a strong indication of diurnal information content in MW LST.

*I am wondering why don't they use MW and IR signal to build diurnal temperature cycle directly. Could this method get more accurate daily global ET? Xuelong Chen has found MODIS monthly LST products could capture the monthly mean of diurnal LST variation. This means that the ALEXI could be used to MODIS monthly LST products. The authors might be interested to the following figure 1.*

» It is certainly possible that we will eventually merge MW and IR to get a better estimate of diurnal temperature. However, this still requires some additional steps to overcome, not least of which is the

spatial resolution difference. This paper presents a milestone along the way, reporting on an experiment to test MW LST in an ET retrieval that is itself sensitive to the diurnal information.

*Eq1. If possible, please give the equation of DTC3, then the readers could quickly understand what kind of curves were used to fit the diurnal cycle.*

» We added the description of the DTC model as a model that " combines a cosine and an exponential term to describe the effect of the sun and the decrease of surface temperature at night". We think that this level of detail is sufficient for understanding the present paper, and listing the exact equations might distract from the main analysis. For a detailed description, the reader is referred to Holmes et al 2015 and references therein.

Section 2.3.2 now includes this passage: "The DTC model combines a cosine and an exponential term to describe the effect of the sun and the decrease of surface temperature at night and is based on Göttsche and Olesen (2001) with slight adaptations to limit the number of parameters. This implementation (DTC3) is fully described in Holmes et al. (2015). "

*Fig. 4 please specify t1 and t2 time for dTrad.*

» We added to the caption: dTrad is defined in the ALEXI framework as the temperature rise between 1.5 hr after sunrise to 1.5 hr before noon (see Section 2.1).

*Fig. 8 how did the authors cope with different spatial resolution when they calculate Pearson's correlation. ALEXI-IR is 0.05 deg and ALEXI-MW is 0.25 deg. The correlation is at 0.05 deg resolution?*

» The correlation in 3-month anomalies (Fig 8) is calculated based on 0.25 degree resolution data. This is now specified in the caption.

*Fig. 9 why not calculate monthly anomaly? MW provide the possibility of daily ET. 3 months anomaly will provide more consistent spatial patterns. But the performance at daily or monthly is more interesting. Fig. 7, and 8 also have the same question.*

» 3-month anomalies were chosen to represent the larger seasonal patterns in interannual variation. Higher temporal resolution is tested with in situ data's. As can be seen in the below figures, the correlations are rather similar between the 1-month and 3-month anomalies.

[Figure]

Figure 1: Pearson's correlation between anomaly in 1-month (left) and 3-month (right) ET totals as estimated by ALEXI-IR and ALEXI-MW, calculated at 0.25 degree resolution. White areas have no data, grey areas are masked because the standard deviation in in anomaly was below 2.5 mm/month in both estimates.

*Figure. 10, when you calculate pearson correlation between satellite data and Fluxnet observations, daily, weekly, or monthly time series data was used? Please give RMSE at mm/day. This is more comparable to other's result.*

» Thanks for pointing this out, the caption did not specify that these statistics are based on weekly ET values. For ALEXI these weekly values are interpolated from clearsky observations as detailed in Section 2.4. The Fig 10 caption is changed to start with: "Comparison of Pearson correlation (rho) and RMSE between estimates of weekly ET from satellite data and Fluxnet observations."

*Describe what is ti in Equation 1.*

» The definition of t is given in the preceding sentence, 'ti' is the observation time of the temperature measurement. The index 'i' is used to represent the individual observations within a day.

*Table 2 Comparison is based on weekly averages in the period of 2003 to 2011. Why not use daily ET with gaps to calculate R, RMS? This is more useful for the readers to compare other ET products. Surely, weekly averages will give a higher R and low RMS. But MW provide ALEXI with the possibility for daily ET calculation. AND page 2, 'generated a data record of weekly ET' why not daily? as above comments \*

» While the goal of this work is to eventually reduce the temporal interval of consistent ET retrievals to daily, at this moment we followed the existing global ALEXI protocol that allows for evaluation of weekly ET totals. We do look forward to analyzing daily output in future studies, but it was not an option at this stage of the development.

*Table 1, MOD43C3 doesn't have gaps? How did they fill albedo gaps? Please specify at what time step (00:00, 06:00: : :) lapse rate profile is used.*

» In fact, it should have read MCD43B3 and this is a 16-day product. The updated reference (Schaaf et al. 2002) explains: ".. BRDF parameters are produced (via either full or magnitude inversions) for every land or coastal area which is viewed (and atmospherically corrected) at least once over a 16-day period. Land areas that remain completely cloud covered over this period are designated with fill values."

[revised manuscript text omitted]

---

## Author Response (AR2)

Dear Editor,

The manuscript was changed to reflect the remaining minor concerns of reviewer 1. Please find below the point-by-point reply (in italic) to the review report.

On behalf of all authors,

Thomas Holmes

Reviewer 1 report.

The paper has been improved and the authors have addressed most of my concerns by better explaining in the new manuscript the conflicting or unclear statements. The more comprehensive roadmap at the end also signals more clearly the limitations of the present exercise, and remove some of my concerns. In my opinion, the paper is ready for publication.

A couple of very minor things, that could even be addressed in the editing phase, no need for a new revision:

P4. L6. I think this is not clear: " In practice this means taking an average of all needed inputs on the "clear-sky" days in the 7-day period and running ALEXI. Based on the resulting "clear sky" ET, we calculate a ratio of instantaneous latent heat flux to incoming solar radiation (fSUN). This fSUN is held constant over the 7-day period and used to calculate daily ET on each day, which is then summed to a 7-day total ET".

I think you have not stated the "instantaneous" calculation of ALEXIS at noon time before. Much better this from one of your answers to the reviewers: "The ALEXI model computes the energy balance at two instantaneous points during the morning hours (post-dawn and pre-noon) using LST data available at those times. The latent heat estimate at the second time is then upscaled to a daily flux, conserving a flux ratio metric." Please, rephrase to make it clear to non-ALEXIS experts would be nice.

*Reply. We rearranged the information of this paragraph in Section 2.1 and rephrased key sentences according to the reviewer suggestions to: "… This initial global ALEXI implementation differs from prior geostationary implementations in that its analysis is performed at weekly timescales. While a daily system is in preparation, at present, the global model is executed using 7-day averages of all inputs on "clear-sky" days to minimize computational load. In practice this means taking an average of all needed inputs (at time 1 and 2) on the "clear-sky" days in the 7-day period and running ALEXI. As in prior geostationary implementations the retrieved latent heat estimate at time 2 is upscaled to a daily flux, conserving a flux ratio metric and using daily solar radiation retrievals. This accounts for changes in atmospheric demand while preserving the scaling flux ratio as determined on the clear-sky days. However, because the scaling flux ratio is held constant over the 7-day period the output is also reported as 7-day total ET (mm/week). …."*

Reply to original P3.L11 comment. You say: "However, we prefer that ALEXI remains a purely diagnostic model to maintain its role as independent estimate of ET, not requiring precipitation data as input". I wonder what makes you claim that not using precipitation makes your model more "diagnostic" and

"independent" than a model that uses precipitation (e.g., GLEAM). Satellite-based precipitation products exist, so ingesting that should not make the model less "diagnostic". Also, in principle, there is nothing making them more "dependent" of whatever you may have in mind, compared with other products such as radiation. For instance, radiation at the surface typically needs an atmospheric model and atmospheric inputs to remove the atmospheric component, where the inputs are typically outsourced from atmospheric reanalysis. So their "independence" may also be questioned. Perhaps, you refer to evaluate ALEXIS ET later on with correlated hydrological datasets such as precipitation, or soil moisture.

*Reply. In the particular case of canopy interception there is indeed a rather direct link between rainfall and subsequent evaporation of the water on the leaves, and our worries about making ALEXI 'less diagnostic' are perhaps misplaced. In the context of the available global ET models we do indeed think that using LST to 'diagnose' the flux partitioning makes ALEXI a uniquely valuable asset to further constrain uncertainty in global ET estimates exactly because it is independent from alternative formulations that use precipitation as the main driver.*

P6.L4. "This overlap in coverage" may be unclear. There is also overlap between MSG and GCOM-W, but the key is that it is not global. Better to say again that the calibration with MODIS is preferred as the present calibration with MSG is not global.

*Reply. Thanks for pointing this out so that we can make the meaning clearer. In section 2.3.5 we now state: "The multi-year and global record of simultaneous MW-LST and MODIS LST during clear skies can support further calibration of MW-LST to MODIS LST in future investigations. This MW to MODIS calibration was not done in this study but is likely needed to maximize consistency between ALEXI implementations over the globe."*

Reply to original P11. L15 comment. This does not look like a serious answer. I think you should explain why we should be surprised or not, and it seems that you were surprised. Perhaps ALEXIS is very sensitive to small change in LST, so you were expecting a worst agreement between ALEXIS MW and IR given the differences between both LTS, even if the LST differences were small. Or perhaps the MW and IR LST estimates were closer than you expected, so even if ALEXIs is not very robust to LST uncertainties, the ALEXIS MW and IR ET are closer than expected.

*Reply. Apologies if that reply came across as too short. The context of this study as a continental scale validation of the diurnal aspect of the MW-LST versus previous studies that only looked at limited numbers of flux tower sites was earlier explained in the answer to the original similar comment on the first page (P1. L11). "In prior papers, we looked at temporal agreements between the satellite estimates and ground measurements from flux tower sites. What the analysis in this paper adds is a continental scale validation of the MW LST diurnal information using MODIS TIR LST. While the reviewer is correct that if the new LST inputs are reasonable, the ALEXI ET retrievals be reasonable, we move to the ET stage here to demonstrate sensitivity of this particular model to expected errors in the diurnal LST curve. The results show that the ET retrievals have some spatial differences that are explainable, but overall the TIR and MW ET estimates are sufficiently consistent under clear-sky conditions such that the next phases of investigation are warranted; namely evaluation of all-sky MW performance and development of a merged TIR-MW product."*

[revised manuscript text omitted]